# Hölder++: Improving the Quality-Coherence Trade-off in Multimodal VAEs

**Huyen Vo** [1 2]   **María Martínez-García** [1]   **Isabel Valera** [1 2]

## Abstract

Existing approaches for multimodal variational autoencoders (VAEs) face a trade-off between generative quality and coherence—i.e., they struggle to generate realistic and diverse samples that, at the same time, are semantically consistent across modalities. A recent work shows that using a simple approximation to Hölder pooling as an aggregation method improves coherence over the SOTA MMVAE+, despite assuming a single shared representation across all modalities. Yet, it slightly compromises sample diversity. Inspired by this insight, we propose Hölder++, a novel multimodal VAE that improves the generative quality-coherence trade-off through: (i) the first implementation of *Hölder pooling without any approximation* for multimodal VAEs; (ii) an extended architecture that models *distinct shared and private* (i.e., modality-specific) representations (Hölder+); and (iii) *hierarchical inference* that further enhances the disentanglement between the shared and private representations (Hölder++). Our experiments corroborate that Hölder++ consistently improves the generative quality-coherence trade-off, yields more structured latent spaces, and learns shared representations that are informative for downstream tasks.

## 1. Introduction

Multimodal variational autoencoders (VAEs) are commonly designed and evaluated along two, often competing, criteria: *generative quality*, reflecting realism and sample diversity; and *generative coherence*, capturing semantic consistency across modalities. A key modeling design choice influencing the *generative quality-coherence trade-off* is the aggregation mechanism, which combines the representations produced by unimodal encoders into a shared representation across all modalities. The predominant approaches in the literature are Product-of-Experts (PoE) (Wu & Goodman, 2018) and Mixture-of-Experts (MoE) (Shi et al., 2019), which unfortunately suffer, respectively, from low coherence or reduced sample diversity (Daunhawer et al., 2022).

To address these limitations, Palumbo et al. (2023) introduced a new training objective that explicitly learns distinct shared and private (modality-specific) latent representations while avoiding *shortcuts*, resulting in MMVAE+. This was the first approach to achieve strong performance on both criteria, demonstrating that structuring the latent space is indeed crucial for improving the quality-coherence trade-off. A different line of work has focused on improving generative coherence by enforcing explicit disentanglement between private and shared subspaces through auxiliary loss terms, often grounded in information bottleneck or mutual information principles (Zhang et al., 2026; Gao et al., 2026; Lee & Pavlovic, 2020; Daunhawer et al., 2020).

Orthogonal to these works, Vo & Valera (2026) recently showed that the quality-coherence trade-off can also be improved through the choice of the aggregation method. In particular, they demonstrated that PoE and MoE can be viewed as special cases of Hölder pooling, a probabilistic opinion pooling framework that minimizes the $\alpha$-divergence between the aggregated and individual densities. Focusing on the symmetric case $\alpha = 0.5$, they proposed a simple moment-matching approximation, Hellinger aggregation, which leads to an improved balance between generative quality and coherence. Notably, this approach achieves higher coherence than MMVAE+ despite relying on a single shared representation, at a slight deterioration of diversity.

**Contributions.** Motivated by these insights, we propose **Hölder++**, *a novel multimodal VAE that improves the state-of-the-art (SOTA) generative quality-coherence trade-off through symmetric Hölder pooling (i.e., $\alpha = 0.5$) and two key architectural choices.* Specifically, our contributions are threefold: (i) we provide the first implementation of symmetric Hölder pooling ($\alpha = 0.5$) as the aggregation mechanism for multimodal VAEs, without relying on approximations; (ii) we extend this framework with distinct shared and private latent subspaces, yielding the Hölder+ model; and (iii) we introduce top-down hierarchical inference to

[1]Department of Computer Science, Saarland University, DE [2]MPI-SWS, Saarland Informatics Campus, DE. Correspondence to: Huyen Vo <vothuckhanhhuyenvn@gmail.com>.

*Proceedings of the 43rd International Conference on Machine Learning*, Seoul, South Korea. PMLR 306, 2026. Copyright 2026 by the author(s).

enforce disentanglement between shared and private representations by design, resulting in Hölder++. Experiments on four benchmark datasets (PolyMNIST, MNIST-SVHN, CUBICC, and CelebAMask-HQ) show that Hölder++ consistently improves the quality-coherence trade-off, learns more structured and disentangled latent spaces, and infers shared representations informative for downstream tasks.

## 2. Background

### 2.1. Multimodal VAEs

Given data $\boldsymbol{X}$ consisting of $M$ modalities, $\boldsymbol{X} := \{\boldsymbol{x}_1, \boldsymbol{x}_2, \ldots, \boldsymbol{x}_M\}$, multimodal VAEs define a generative model with a shared latent variable $\boldsymbol{z}$. Under the assumption that modalities are conditionally independent given $\boldsymbol{z}$, the joint generative distribution factorizes as $p(\boldsymbol{X}, \boldsymbol{z}) = p(\boldsymbol{z}) \prod_{j=1}^{M} p_{\boldsymbol{\theta}_j}(\boldsymbol{x}_j | \boldsymbol{z})$, where $p(\boldsymbol{z})$ denotes the prior over the shared latent space and $p_{\boldsymbol{\theta}_j}(\boldsymbol{x}_j | \boldsymbol{z})$ is a modality-specific likelihood parameterized by a neural decoder with parameters $\boldsymbol{\theta}_j$. Multimodal VAEs are often trained by maximizing the Evidence Lower Bound (ELBO), given by

$$\text{ELBO} = \mathbb{E}_{q_\Phi(\boldsymbol{z}|\boldsymbol{X})}[\log p_{\boldsymbol{\theta}}(\boldsymbol{X}|\boldsymbol{z})] - \text{KL}(q_\Phi(\boldsymbol{z}|\boldsymbol{X})\|p(\boldsymbol{z})),$$

where the distribution $q_\Phi(\boldsymbol{z}|\boldsymbol{X})$ denotes a variational posterior with learnable parameters $\Phi$. In this framework, the PoE (Wu & Goodman, 2018) and MoE (Shi et al., 2019) are standard approaches to obtain joint posterior approximations that scale with the number of modalities. The PoE approximates the joint posterior as $q_\Phi(\boldsymbol{z}|\boldsymbol{X}) = c \prod_{j=1}^{M} q_{\boldsymbol{\phi}_j}(\boldsymbol{z}|\boldsymbol{x}_j)$, where $c$ is a normalization constant that ensures a valid probability distribution, whereas the MoE models it as an equally weighted mixture, given by $q_\Phi(\boldsymbol{z}|\boldsymbol{X}) = \frac{1}{M} \sum_{j=1}^{M} q_{\boldsymbol{\phi}_j}(\boldsymbol{z}|\boldsymbol{x}_j)$. In both cases, the joint posterior is constructed from modality-specific neural encoders with parameters $\boldsymbol{\phi}_j$, where $j \in \{1, 2, \ldots, M\}$.

Recent works have proposed novel aggregation methods for approximating $q_\Phi(\boldsymbol{z}|\boldsymbol{X})$ in this setting, including CoDE-VAE (Mancisidor et al., 2025), which leverages a consensus of dependent experts; WBVAE (Qiu et al., 2025), which adopts the 2-Wasserstein barycenter; and HELVAE (Vo & Valera, 2026), which introduces Hellinger aggregation, a Laplace approximation to Hölder pooling with $\alpha = 0.5$.

### 2.2. Hölder pooling

Probabilistic opinion pooling provides a principled approach for aggregating multiple probability density functions $\{q_j(\boldsymbol{z})\}_{j=1}^{M} \in \mathcal{P}^M$ into a single consensus (pooled) distribution as a weighted aggregation of individual densities, where the non-negative weights $\{\lambda_j\}_{j=1}^{M}$ satisfy $\sum_{j=1}^{M} \lambda_j = 1$. The pooling function is obtained by minimizing a weighted average of a chosen discrepancy measure between the aggregated and individual densities. Con-

sidering the family of $\alpha$-divergences, this corresponds to $q(\boldsymbol{z}) = \arg \min_{\varphi \in \mathcal{P}} \sum_{j=1}^{M} \lambda_j \mathcal{D}_\alpha(q_j \| \varphi)$, which yields an $\alpha$-parameterized family of Hölder pooling functions (Garg et al., 2004; Koliander et al., 2022). Recent work has studied PoE and MoE as special cases of Hölder pooling (Vo & Valera, 2026) and, based on this insight, proposed a novel aggregation method considering the symmetric case of the $\alpha$-divergence family, which corresponds to $\alpha = 0.5$ (Hernandez-Lobato et al., 2016). Since learnable weights can degrade quality and coherence by allowing a few modalities to dominate (Vo & Valera, 2026), we follow PoE and MoE and use uniform pooling weights. The resulting aggregated distribution is therefore given as follows:

$$q(\boldsymbol{z}) = c \left( \sum_{j=1}^{M} q_j(\boldsymbol{z}) + 2 \sum_{i=1}^{M} \sum_{j>i}^{M} \sqrt{q_i(\boldsymbol{z})q_j(\boldsymbol{z})} \right), \quad (1)$$

where $c = 1/\int \left( \sum_{j=1}^{M} \sqrt{q_j(\boldsymbol{z})} \right)^2 d\boldsymbol{z}$. Moreover, Vo & Valera (2026) derived a Laplace approximation to the symmetric Hölder pooling posterior via moment matching. The resulting method, referred to as Hellinger aggregation, significantly improves SOTA performance and achieves a better quality-coherence trade-off in multimodal VAEs that rely on a single latent space shared across all modalities.

### 2.3. Introducing shared and modality-specific subspaces

While earlier works explored shared and modality-specific latent subspaces in multimodal VAEs (Wang et al., 2016; Bouchacourt et al., 2018; Tsai et al., 2019), MM-VAE+ (Palumbo et al., 2023) was the first to achieve strong performance in both generative quality and generative coherence, and it remains a leading SOTA approach. In this framework, modality $\boldsymbol{x}_m$ is modeled with both a *private* (a.k.a. modality-specific) latent $\boldsymbol{w}_m$, and a *shared latent representation* $\boldsymbol{z}$ modeling the shared information across all modalities. The generative model assumes independence between the shared and private representations, factorizing as $p_\Theta(\boldsymbol{X}, \boldsymbol{z}, \boldsymbol{W}) = p(\boldsymbol{z}) \prod_{j=1}^{M} p_{\boldsymbol{\theta}_j}(\boldsymbol{x}_j | \boldsymbol{z}, \boldsymbol{w}_j) p(\boldsymbol{w}_j)$, where $\boldsymbol{W} := \{\boldsymbol{w}_1, \boldsymbol{w}_2, ..., \boldsymbol{w}_M\}$, and the variational posterior is assumed to factorize accordingly as $q_\Phi(\boldsymbol{z}, \boldsymbol{W}|\boldsymbol{X}) = q_{\Phi_{\boldsymbol{z}}}(\boldsymbol{z}|\boldsymbol{X}) \prod_{j=1}^{M} q_{\boldsymbol{\phi}_j}(\boldsymbol{w}_j|\boldsymbol{x}_j)$, where $q_{\Phi_{\boldsymbol{z}}}(\boldsymbol{z}|\boldsymbol{X})$ is approximated using MoE as aggregation method.

Wang et al. (2016), however, showed that such an approach can lead to *shortcuts*, where the modality-specific subspaces capture all the information, thus neglecting the shared representation. To avoid this behavior, Palumbo et al. (2023) introduced a modified objective (see Eq. (8), Appendix A.1) that distinguishes between self- and cross-reconstruction. For each modality, the log-likelihood $\log p(\boldsymbol{x}_n | \boldsymbol{z}, \boldsymbol{w}_n)$ (and thus, the reconstruction loss) is computed using a shared representation $\boldsymbol{z}$ sampled from one of the unimodal encoders, i.e., $\boldsymbol{z} \sim q_{\boldsymbol{\phi}_{\boldsymbol{z}_j}}(\boldsymbol{z}|\boldsymbol{x}_j)$, together with the corresponding pri-

vate latent $\boldsymbol{w}_n$ sampled as:

$$\boldsymbol{w}_n \sim \begin{cases} q_{\boldsymbol{\phi}_{\boldsymbol{w}_n}}(\boldsymbol{w}_n \mid \boldsymbol{x}_n), & n = j \quad \text{(self-term),} \\ r_n(\boldsymbol{w}_n), & n \neq j \quad \text{(cross-term),} \end{cases} \quad (2)$$

where $\{r_n(\boldsymbol{w}_n)\}_{n=1}^M$ are (non-informative) auxiliary prior distributions on the private representations. This design forces the decoder to rely on the shared latent variable $\boldsymbol{z}$ when reconstructing unobserved modalities, thus preventing *shortcuts*. Moreover, CMVAE (Palumbo et al., 2024) extends this approach with a mixture prior over latent $\boldsymbol{z}$ to further enforce structure in the shared latent space.

To improve generative coherence, several recent works enhance disentanglement between shared and private representations via auxiliary losses (Zhang et al., 2026; Gao et al., 2026; Lee & Pavlovic, 2020; Daunhawer et al., 2020). For instance, DCMEM (Gao et al., 2026) relies on a contrastive mutual-information loss, but is limited to bimodal settings. DMVAE (Lee & Pavlovic, 2020) applies total-correlation regularization over the concatenated $[\boldsymbol{z}, \boldsymbol{w}]$ to enforce independence across latent dimensions. Both require nontrivial hyperparameter tuning. In contrast, we propose a variational factorization that is directly applicable to any number of modalities and encourages disentanglement by design.

## 3. Hölder++ VAE

In this section, we present the core components of our method. We first introduce exact symmetric Hölder pooling ($\alpha = 0.5$) as aggregation in multimodal VAEs (Section 3.1). We then incorporate a shared-private latent structure, yielding Hölder+ (Section 3.2). Finally, we introduce hierarchical inference to improve disentanglement between shared and private representations, yielding **Hölder++** (Section 3.3).

### 3.1. Exact (symmetric) Hölder pooling as aggregation

In multimodal VAEs with a single latent space shared across modalities, the Hellinger VAE (HELVAE) (Vo & Valera, 2026) has been shown empirically to improve the quality-coherence trade-off by using Hellinger aggregation, a Laplace approximation of Hölder pooling with $\alpha = 0.5$. In this paper, we *provide the first exact implementation of symmetric Hölder pooling ($\alpha = 0.5$) for multimodal VAEs*. The resulting pooled posterior admits a mixture representation consisting of unimodal and pairwise components, thus explicitly capturing the multimodal nature of the task.

More in detail, when applied to the joint posterior approximation, the Hölder pooling operator in Eq. (1) can be expressed as a mixture of Gaussians, comprising both unimodal and pairwise components, i.e.:

$$q(\boldsymbol{z}|\boldsymbol{X}) = \sum_{j=1}^M \pi_j q_{\boldsymbol{\phi}_{\boldsymbol{z}_j}}(\boldsymbol{z}|\boldsymbol{x}_j) + \sum_{i=1}^M \sum_{j>i}^M \pi_{ij} q_{ij}^{(1/2)}(\boldsymbol{z}|\boldsymbol{x}_i, \boldsymbol{x}_j),$$

where $\pi_j$ and $\pi_{ij}$ denote the mixture weights of, respectively, the unimodal $q_{\boldsymbol{\phi}_{\boldsymbol{z}_j}}(\boldsymbol{z}|\boldsymbol{x}_j)$ and the pairwise $q_{ij}^{(1/2)}(\boldsymbol{z}|\boldsymbol{x}_i, \boldsymbol{x}_j)$ components. For brevity, we omit the explicit dependence on the variational parameters and use the subscript $ij$ for the pairwise components. We remark that the above formulation generalizes MMVAE (Shi et al., 2019), which uses MoE, by including additional pairwise components. Below, we provide the details on how each of these terms is computed; Appendix A.1 contains the full derivations.

The *unimodal mixture components* $\{q_{\boldsymbol{\phi}_{\boldsymbol{z}_j}}(\boldsymbol{z}|\boldsymbol{x}_j)\}_{j=1}^M$ are assumed to be Gaussian probability densities with diagonal covariance in $\mathbb{R}^D$, $\{\mathcal{N}(\boldsymbol{\mu}_j, \mathrm{diag}(\boldsymbol{\sigma}_j^2))\}_{j=1}^M$, with $\boldsymbol{z} \in \mathbb{R}^D$ being the latent representation (i.e., the latent variable), $\boldsymbol{\mu}_j = (\mu_{j,1}, \mu_{j,2}, \ldots, \mu_{j,D})^\top \in \mathbb{R}^D$ the mean vector, and $\boldsymbol{\sigma}_j^2 = (\sigma_{j,1}^2, \sigma_{j,2}^2, \ldots, \sigma_{j,D}^2)^\top \in \mathbb{R}^D$ the variance vector.

Each *pairwise mixture component* $q_{ij}^{(1/2)}(\boldsymbol{z}|\boldsymbol{x}_i, \boldsymbol{x}_j)$ is obtained by normalizing the geometric mean of the corresponding unimodal posteriors $q_{\boldsymbol{\phi}_{\boldsymbol{z}_i}}(\boldsymbol{z}|\boldsymbol{x}_i)$ and $q_{\boldsymbol{\phi}_{\boldsymbol{z}_j}}(\boldsymbol{z}|\boldsymbol{x}_j)$. This results in a Gaussian distribution of the form $q_{ij}^{(1/2)}(\boldsymbol{z}|\boldsymbol{x}_i, \boldsymbol{x}_j) = \mathcal{N}(\boldsymbol{z}; \boldsymbol{\mu}_{ij}, \boldsymbol{\sigma}_{ij}^2)$. The parameters for each modality pair $(i, j)$ with $1 \leq i < j \leq M$ and for each latent dimension $d \in \{1, 2, \ldots, D\}$ are given by:

$$\mu_{ij,d} = \frac{\mu_{i,d}\,\sigma_{j,d}^2 + \mu_{j,d}\,\sigma_{i,d}^2}{\sigma_{i,d}^2 + \sigma_{j,d}^2}, \quad \sigma_{ij,d}^2 = \frac{2\sigma_{i,d}^2 \sigma_{j,d}^2}{\sigma_{i,d}^2 + \sigma_{j,d}^2}.$$

Finally, the *mixture weights* can be computed as $\pi_j = c$ and $\pi_{ij} = 2cS_{ij}$, where $c$ denotes the normalization constant of the aggregated distribution in Eq. (1) and can be computed in closed form as

$$c = \left( M + 2\sum_{i=1}^M \sum_{j>i}^M S_{ij} \right)^{-1};$$

being $S_{ij}$ the Bhattacharyya coefficient between the unimodal posteriors $q_{\boldsymbol{\phi}_{\boldsymbol{z}_i}}$ and $q_{\boldsymbol{\phi}_{\boldsymbol{z}_j}}$, i.e.,

$$S_{ij} = \prod_{d=1}^D \sqrt{\frac{2\sigma_{i,d}\sigma_{j,d}}{\sigma_{i,d}^2 + \sigma_{j,d}^2}} \exp\left( -\frac{(\mu_{i,d} - \mu_{j,d})^2}{4(\sigma_{i,d}^2 + \sigma_{j,d}^2)} \right).$$

**Remark.** Hölder VAE suffers from two limitations relative to HELVAE, both stemming from its mixture subsampling scheme. First, it substantially increases computational complexity by requiring sampling from a mixture with $M^2$ components (see Table 8 in Appendix C.1 for a computational comparison). Second, we expect the Hölder VAE to suffer from limited generative quality, like other mixture-based multimodal VAEs, such as MMVAE (Shi et al., 2019) and MoPoE (Sutter et al., 2021), that rely on mixture subsampling of a single shared representation (Daunhawer et al., 2022) (see results in Section 4.1). Fortunately, as shown in

MMVAE+ (Palumbo et al., 2023), this effect can be mitigated by splitting the latent representation into shared and private subspaces. Thus, motivating our Hölder+ extension.

### 3.2. Hölder+: Learning shared and private subspaces

Models that rely on a single latent space shared across modalities often empirically suffer from limited sample diversity, a phenomenon observed in novel methods such as HELVAE (Vo & Valera, 2026) or CoDEVAE (Mancisidor et al., 2025), despite the improvements in generative coherence. We address this limitation by factorizing the latent space into shared and modality-specific subspaces. To prevent *shortcuts*, where private latent representations capture shared semantic information, we adopt the scheme proposed in MMVAE+ and rely on auxiliary non-informative distributions to sample private features in the cross-modal reconstruction terms (Palumbo et al., 2023). As highlighted in the previous section, the latter design choice forces the decoder to rely only on the shared latent $z$ when reconstructing unobserved modalities, thus preventing shortcuts.

Specifically, when $z$ is sampled from a unimodal component $j$, i.e., $z \sim q_{\phi_j}(z|x_j)$, we compute the conditional log-likelihood terms $\log p(x_n|z, w_n)$ for all modalities $n \in \{1, 2, \ldots, M\}$, where each $w_n$ is sampled following Eq. (2). Analogously, when $z$ is sampled from a pairwise component $(i, j)$, i.e., $z \sim q_{ij}^{(1/2)}(z|x_i, x_j)$, we evaluate the same reconstruction terms with $w_n$ sampled as

$$
w_n \sim \begin{cases} q_{\phi_{w_n}}(w_n|x_n), & n \in \{i, j\}, \\ r_n(w_n), & n \notin \{i, j\}, \end{cases} \quad (3)
$$

where $\{r_n(w_n)\}_{n=1}^{M}$ are auxiliary, non-informative priors for the private (i.e., modality-specific) latent representations. We refer to the resulting model as **Hölder+**, with the full objective presented in Appendix A.1, Eq. (9). We remark that Hölder+ optimizes a valid ELBO and is therefore a proper multimodal VAE, as proven in Appendix A.2.

**Remark.** We stress that, while HELVAE shows advantages over Hölder in a single shared representation setting, that is not the case when considering distinct shared-private subspaces. This is due to the fact that Hellinger aggregation does sample the shared representation $z$ after aggregating all the modalities, thus failing to distinguish from self- and cross-reconstruction (and sampling) (see Eqs. (2) and (3)), which is essential to avoid shortcuts. Consequently, we expect Hölder+ to yield improved quality-coherence trade-offs relative to HELVAE, analogously to the improvement of MMVAE+ over MMVAE (see experiments in Section 4.1).

### 3.3. Hölder++: Disentangling private-shared subspaces

Existing methods promote shared-private disentanglement by introducing auxiliary loss terms based on information-

bottleneck or mutual-information principles (Zhang et al., 2026; Gao et al., 2026; Lee & Pavlovic, 2020; Daunhawer et al., 2020). In contrast, we propose a variational factorization that encourages *disentanglement by design*.

Multimodal VAEs often approximate the posterior distribution by assuming that shared and private representations are conditionally independent given the data, i.e., $q_\Phi(z, W|X) = q_{\phi_z}(z|X) q_{\phi_W}(W|X)$. However, the true posterior generally does not factorize with respect to either the shared or the private representations, even if independence is assumed in the prior. The variational factorization should therefore be understood as a deliberate modeling assumption introduced to obtain a tractable approximation of the true posterior. To improve this approximation, we adopt hierarchical inference by parameterizing the variational posterior as

$$
q_\Phi(z, W|X) = q_{\Phi_z}(z|X) \prod_{j=1}^{M} q_{\phi_{w_j}}(w_j|x_j, z),
$$

i.e., we first infer the shared latent representation $z$ (the *top level* of the hierarchy), which captures semantics shared across all modalities, and then infer each modality-specific latent $w_j$ (the *bottom level*) conditioned on both the modality input $x_j$ and the shared latent $z$. This design choice reflects the intuition that, in multimodal VAEs, generative coherence across modalities can only be achieved with an informative shared representation. Assuming that the shared and private representations act as information bottlenecks, adopting a top-down factorization of the approximate posterior introduces an effective inductive bias that prevents $w_j$ from capturing all the information in data (Sønderby et al., 2016; Vahdat & Kautz, 2020; Havtorn et al., 2021). Instead, $\{w_j\}_{j=1}^{M}$ models residual modality-specific information in $x_j$ that is not already captured by $z$, thereby avoiding undesirable shortcuts in practice.

By applying hierarchical inference to Hölder+, we obtain **Hölder++**. Figure 1 shows the Hölder++ graphical model for the unimodal and pairwise components (panels (a) and (b)) and the corresponding training objective (panel (c)), highlighting the modifications relative to Hölder+.

**Remark.** Our approach is fundamentally different from the Hierarchical Multimodal VAE (HMVAE) (Wolff et al., 2022), which uses a top-down hierarchy for both inference and generation. In HMVAE, each modality-specific latent is conditioned on the shared latent, and only the private representations are passed to the unimodal decoders, potentially limiting the coherence across modalities if the modality-specific representations are expressive enough to capture all the information in the data. In our implementation of Hölder++, we instead assume that the private and shared representations are independent *a priori*, and use hierarchical inference to enhance disentanglement during inference.

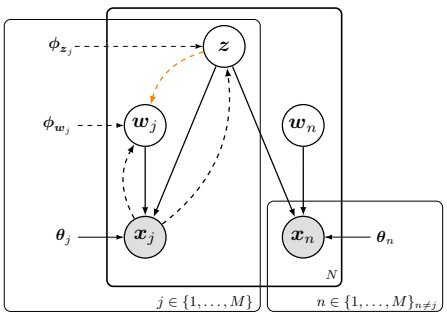

*(a)* Unimodal components in the Hölder++ objective

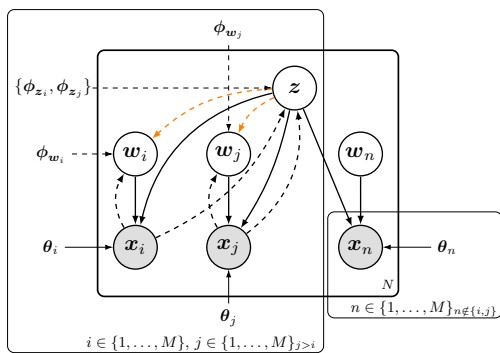

*(b)* Pairwise components in the Hölder++ objective

$$\mathcal{L}_{\text{Hölder++}}(\boldsymbol{x}_{1:M}) = \sum_{j=1}^{M} \pi_j \mathbb{E}_{\substack{q_{\boldsymbol{\phi}_{\boldsymbol{z}_j}}(\boldsymbol{z}|\boldsymbol{x}_j) \\ q_{\boldsymbol{\phi}_{\boldsymbol{w}_j}}(\boldsymbol{w}_j|\boldsymbol{x}_j,\boldsymbol{z}) \\ \{\tilde{\boldsymbol{w}}_n \sim r_n(\boldsymbol{w}_n)\}_{n\neq j}}} \log\left( \frac{p_{\boldsymbol{\theta}_j}(\boldsymbol{x}_j|\boldsymbol{z},\boldsymbol{w}_j)p(\boldsymbol{z})p(\boldsymbol{w}_j)}{q_{\boldsymbol{\phi}_{\boldsymbol{z}}}(\boldsymbol{z}|\boldsymbol{X})q_{\boldsymbol{\phi}_{\boldsymbol{w}_j}}(\boldsymbol{w}_j|\boldsymbol{x}_j,\boldsymbol{z})} \prod_{n\neq j} p_{\boldsymbol{\theta}_n}(\boldsymbol{x}_n|\boldsymbol{z},\tilde{\boldsymbol{w}}_n) \right)$$

$$+ \sum_{i=1}^{M}\sum_{j>i}^{M} \pi_{ij} \mathbb{E}_{\substack{q_{ij}^{(1/2)}(\boldsymbol{z}|\boldsymbol{x}_i,\boldsymbol{x}_j) \\ q_{\boldsymbol{\phi}_{\boldsymbol{w}_i}}(\boldsymbol{w}_i|\boldsymbol{x}_i,\boldsymbol{z}) \\ q_{\boldsymbol{\phi}_{\boldsymbol{w}_j}}(\boldsymbol{w}_j|\boldsymbol{x}_j,\boldsymbol{z}) \\ \{\tilde{\boldsymbol{w}}_n \sim r_n(\boldsymbol{w}_n)\}_{n\notin\{i,j\}}}} \log\left( \frac{p_{\boldsymbol{\theta}_i}(\boldsymbol{x}_i|\boldsymbol{z},\boldsymbol{w}_i)p_{\boldsymbol{\theta}_j}(\boldsymbol{x}_j|\boldsymbol{z},\boldsymbol{w}_j)p(\boldsymbol{z})p(\boldsymbol{w}_i)p(\boldsymbol{w}_j)}{q_{\boldsymbol{\phi}_{\boldsymbol{z}}}(\boldsymbol{z}|\boldsymbol{X})q_{\boldsymbol{\phi}_{\boldsymbol{w}_i}}(\boldsymbol{w}_i|\boldsymbol{x}_i,\boldsymbol{z})q_{\boldsymbol{\phi}_{\boldsymbol{w}_j}}(\boldsymbol{w}_j|\boldsymbol{x}_j,\boldsymbol{z})} \prod_{n\notin\{i,j\}} p_{\boldsymbol{\theta}_n}(\boldsymbol{x}_n|\boldsymbol{z},\tilde{\boldsymbol{w}}_n) \right).$$

*(c)* Hölder++ objective

*Figure 1.* A graphical-model view of the unimodal and pairwise components used by Hölder++ (top) and the resulting training objective (bottom). Gray circles denote observed variables, white circles denote latent variables, and non-circled symbols denote model parameters. Solid arrows indicate the generative process, while dashed arrows indicate amortized posterior inference. The objective is a weighted sum of unimodal and pairwise ELBO-style terms, where highlighted terms indicate the modifications introduced by hierarchical inference relative to the baseline architecture. We apply the shortcut-avoiding scheme in Eqs. (2) and (3) to the hierarchical modality-specific posteriors, i.e., $\{q_{\boldsymbol{\phi}_{\boldsymbol{w}_j}}(\boldsymbol{w}_j|\boldsymbol{x}_j,\boldsymbol{z})\}_{j=1}^{M}$, with $j \in \{1, 2, \ldots, M\}$. For clarity, the graphical model explicitly shows that, in the pairwise component $(i,j)$, the posterior over $\boldsymbol{z}$ is parameterized by the unimodal encoders $\boldsymbol{\phi}_{\boldsymbol{z}_i}$ and $\boldsymbol{\phi}_{\boldsymbol{z}_j}$.

Yet, extensions to account for a top-down (from shared to private) generative model of Hölder++ are straightforward and could be of interest particularly in some applications, e.g., where the shared content affects the modality-specific style (Von Kügelgen et al., 2021; Daunhawer et al., 2023).

## 4. Experimental results

**Datasets.** We evaluate our approach on four standard benchmark datasets: PolyMNIST (Sutter et al., 2021), MNIST-SVHN (Shi et al., 2019), CUBICC (Palumbo et al., 2024), and CelebAMask-HQ (Lee et al., 2020). PolyM-NIST is a synthetic dataset with five modalities, where each example is generated by patching MNIST digits of the same class onto random crops from five background images. MNIST-SVHN pairs MNIST and Street View House Numbers (SVHN) digits with the same labels but different visual styles. CUBICC, a variant of the CUB image-caption dataset, contains bird images paired with textual descriptions, grouped into eight species categories, and we use it to evaluate downstream clustering. CelebAMask-HQ is a real-world dataset in which images, masks, and attributes are different modalities describing visual characteristics.

**Baselines.** We compare against SOTA methods that use a single latent shared across modalities—MVAE (Wu & Goodman, 2018), MMVAE (Shi et al., 2019), MoPoE (Sutter et al., 2021), and HELVAE (Vo & Valera, 2026)—as well as approaches that use distinct shared and modality-specific latents, including DMVAE (Lee & Pavlovic, 2020), MMVAE+ (Palumbo et al., 2023), CMVAE (Palumbo et al., 2024), and DCMEM (Gao et al., 2026). For a fair comparison with CMVAE on the downstream clustering task, we also apply the mixture prior on $\boldsymbol{z}$ to Hölder+ and Hölder++, yielding CHölder+ and CHölder++. All experiments report average performance over 3 random seeds, except on CU-BICC, where we use 10 seeds. Further experimental details and results are provided in Appendices B and C.

**Metrics.** We assess the **quality-coherence trade-off** using Fréchet Inception Distance (FID) (Heusel et al., 2017) as a measure of generative quality and classification accuracy on the generated samples as coherence, except for CelebAMask-HQ where we use F1 score. We also evaluate the quality of the posterior approximation using the ELBO.

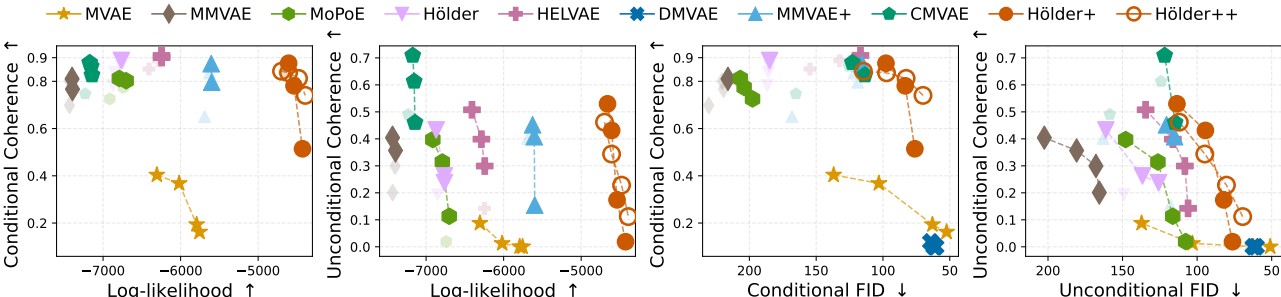

Figure 2. Trade-offs on PolyMNIST between generative coherence (↑) and log-likelihood estimation (↑), as well as between generative coherence (↑) and generative quality (FID ↓), with $\beta \in \{1, 2.5, 5, 10\}$. For each model, the Pareto front (dashed line) connects the non-dominated points that achieve the best trade-offs. Optimal region: upper-right for all plots.

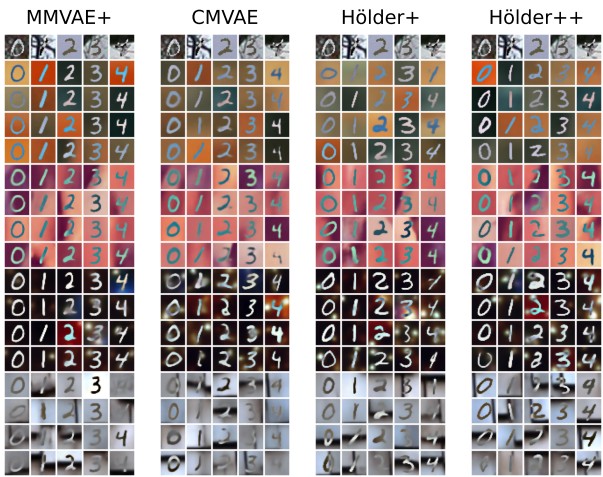

Figure 3. Qualitative results for conditional generation on PolyM-NIST. The input example from the first modality is shown in the top row, and the rows below display four conditional samples for each of the remaining modalities.

To assess the **disentanglement** between shared and private subspaces, we evaluate it both directly on the inferred representations and in the generated images. For the former, we follow prior work by training a linear classifier on the latent spaces and reporting accuracy. We expect high accuracy for the shared latent $z$ and low accuracy for the private latent $w$, which should not encode class information. For the latter, we introduce **three new disentanglement metrics**: $z$ *content stability (↑)*, $z$ *content accuracy (↑)*, and $w$ *content accuracy (↓)*. For the first two, we fix $z$, sample multiple $w \sim p(w)$, decode, and classify the outputs. While $z$ *content stability* measures agreement across samples, $z$ *content accuracy* measures accuracy with respect to the ground-truth label. Higher values indicate that content is captured exclusively by $z$, as fixing $z$ yields outputs with accurate and invariant content despite variations in $w$. For $w$ *content accuracy*, we fix $w$, sample multiple $z \sim p(z)$, decode, classify the outputs, and measure classification accuracy. Lower values reflect stronger disentanglement. All three metrics are averaged over self- and cross-generation.

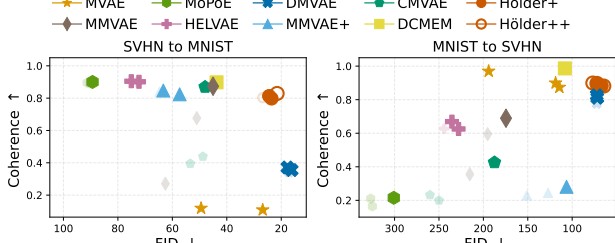

Figure 4. Trade-offs on MNIST-SVHN between conditional generative coherence (↑) and conditional generative quality (FID ↓). For each model, non-dominated points are larger, and dominated points have low opacity. Optimal region: upper-right for both.

Finally, we assess representation quality via **downstream clustering** on the shared latent space using K-means and report accuracy (ACC), normalized mutual information (NMI), and adjusted rand index (ARI).

### 4.1. Generative quality and coherence trade-off

**PolyMNIST.** PolyMNIST has five modalities, so we omit DCMEM, as it is restricted to bimodal settings. We evaluate generative coherence and quality using samples drawn from both the joint posterior (conditional) and the prior (unconditional). Figure 2 summarizes these trade-offs across generative performance metrics for different $\beta$ values (Higgins et al., 2017). Here, we first observe that HELVAE outperforms the rest of the *models with a single shared representation*, including Hölder. Yet, the latter significantly improves the achieved trade-offs compared to MMVAE and MoPoE, showing that the symmetric Hölder pooling improves both quality and coherence, even when limited by mixture subsampling. These results confirm our insights in Section 3.1 and thus argue against the use of mixture sub-sampling in shared-representation models. Consequently, we will not report further Hölder VAEs, but only HELVAE.

Moving to *models with distinct shared-private representations*, we first observe that DMVAE achieves the lowest FID but exhibits poor coherence; in particular, DMVAE's factorized representations are highly prone to shortcut solu-

*Table 1.* Conditional generative coherence (F1 score ↑) and conditional generative quality (FID ↓) on CelebAMask-HQ. The given modalities are used to generate the modality on top. The best and second-best results are marked bold and underlined, respectively.

| | Attribute (F1 ↑) | | Mask (F1 ↑) | | Image (FID ↓) | | |
| | Mask + Image | Image | Attribute + Image | Image | Mask + Attribute | Mask | Attribute |
| --- | --- | --- | --- | --- | --- | --- | --- |
| MMVAE+ | $0.596 \pm 0.016$ | $0.601 \pm 0.026$ | $\mathbf{0.802} \pm \mathbf{0.021}$ | $0.879 \pm 0.006$ | $98.33 \pm 6.83$ | $92.63 \pm 4.67$ | $110.15 \pm 6.81$ |
| CMVAE | $0.590 \pm 0.007$ | $0.609 \pm 0.007$ | $0.798 \pm 0.008$ | $0.874 \pm 0.009$ | $104.92 \pm 6.06$ | $95.91 \pm 1.52$ | $125.21 \pm 8.53$ |
| Hölder+ | $\underline{0.632} \pm 0.003$ | $\underline{0.655} \pm 0.001$ | $\underline{0.800} \pm 0.004$ | $\mathbf{0.896} \pm \mathbf{0.001}$ | $\mathbf{78.82} \pm \mathbf{2.44}$ | $\mathbf{72.32} \pm \mathbf{2.11}$ | $\mathbf{87.19} \pm \mathbf{1.22}$ |
| Hölder++ | $\mathbf{0.633} \pm \mathbf{0.007}$ | $\mathbf{0.665} \pm \mathbf{0.007}$ | $0.792 \pm 0.008$ | $\underline{0.885} \pm 0.017$ | $\underline{80.73} \pm 7.66$ | $\underline{73.64} \pm 6.52$ | $\underline{90.99} \pm 8.66$ |

*Table 2.* Digit classification accuracy on the latent representations for MNIST-SVHN, evaluated on the private $w$, shared $z$, and joint $[w, z]$ subspaces. The best and second-best results are marked bold and underlined, respectively.

| | MNIST Representation | | | SVHN Representation | | |
| | Joint (↑) | Shared (↑) | Private (↓) | Joint (↑) | Shared (↑) | Private (↓) |
| --- | --- | --- | --- | --- | --- | --- |
| DMVAE | $0.970 \pm 0.001$ | $0.862 \pm 0.010$ | $0.673 \pm 0.043$ | $0.900 \pm 0.010$ | $0.898 \pm 0.009$ | $\underline{0.113} \pm 0.001$ |
| MMVAE+ | $0.953 \pm 0.002$ | $0.857 \pm 0.026$ | $0.471 \pm 0.056$ | $0.910 \pm 0.003$ | $0.911 \pm 0.003$ | $0.118 \pm 0.004$ |
| DCMEM | $\mathbf{0.989} \pm \mathbf{0.001}$ | $\mathbf{0.989} \pm \mathbf{0.001}$ | $\mathbf{0.241} \pm \mathbf{0.010}$ | $0.911 \pm 0.002$ | $0.913 \pm 0.004$ | $0.129 \pm 0.002$ |
| CMVAE | $0.948 \pm 0.001$ | $0.861 \pm 0.060$ | $0.389 \pm 0.135$ | $0.913 \pm 0.002$ | $0.914 \pm 0.002$ | $0.116 \pm 0.001$ |
| Hölder+ | $0.976 \pm 0.001$ | $0.966 \pm 0.003$ | $0.479 \pm 0.007$ | $\mathbf{0.923} \pm \mathbf{0.004}$ | $\mathbf{0.922} \pm \mathbf{0.004}$ | $0.114 \pm 0.001$ |
| Hölder++ | $\underline{0.977} \pm 0.001$ | $\underline{0.970} \pm 0.001$ | $\underline{0.387} \pm 0.023$ | $\underline{0.922} \pm 0.001$ | $\underline{0.922} \pm 0.001$ | $\mathbf{0.112} \pm \mathbf{0.001}$ |

tions. Disregarding DMVAE, we observe that shared-private models, in general, outperform their single-representation counterparts in the achievable quality-coherence trade-offs, except for HELVAE, which remains overall competitive. Importantly, *our Hölder+ and Hölder++ yield the best Pareto trade-offs*, combining the highest log-likelihood with competitive FID while matching HELVAE in coherence. Although CMVAE achieves the highest unconditional coherence due to its mixture prior over $z$, it significantly underperforms in log-likelihood. The large improvement of Hölder+ over MMVAE+ shows again the benefit of symmetric Hölder pooling. Moreover, Hölder++ closely matches Hölder+, indicating that hierarchical inference maintains Pareto optimality. Qualitative results in Figure 3 further confirm the generative quality of Hölder+ and Hölder++.

**MNIST-SVHN.** Figure 4 shows the coherence-FID trade-off for conditional cross-modal generation (across $\beta$ values). Several baselines exhibit a clear gap between directions: MMVAE+ and CMVAE perform well for SVHN→MNIST but degrade substantially for MNIST→SVHN, whereas MVAE and DMVAE show the opposite trend. In contrast, *Hölder+ and Hölder++ remain in the upper-right region in both directions*, combining consistently high coherence with among the overall lowest FID. Relative to DCMEM (varying $\alpha$), Hölder-based models deliver higher sample quality (lower FID) while being competitive in coherence. Finally, as before, Hölder++ overlaps Hölder+.

**CelebAMask-HQ.** Table 1 reports F1 scores for attribute and mask prediction, and FID for image generation, conditioned on subsets of the remaining modalities. Compared to MMVAE+ and CMVAE, Hölder+ and Hölder++ consistently achieve the best or second-best results across nearly all conditioning settings, with Hölder+ achieving the high-

est image quality and Hölder++ yielding the best attribute prediction. These results confirm that *our methods improve quality-coherence trade-offs on a challenging real-world benchmark*. To further enhance visual fidelity, we apply a diffusion model as a post-hoc refinement step. Figure 12 in Appendix C.7 shows that feeding Hölder++ samples into a pretrained DiffuseVAE (Pandey et al., 2024) improves image quality without changing the underlying characteristics.

**Take-away.** *While HELVAE remains the SOTA among single shared-representation models, Hölder+ and Hölder++ consistently achieve the best Pareto frontier* compared to all competing methods overall. Moreover, Hölder++ closely matches Hölder+, showing that hierarchical inference preserves the coherence-quality trade-off. This improvement comes at the cost of increased computational complexity due to the pairwise terms in our objective, which also contribute to the gains of Hölder+/++ over SOTA models. However, we do not observe a quadratic increase in training time as the number of modalities grows (see Appendix C.1 for details).

### 4.2. Disentanglement of shared and private subspaces

For PolyMNIST, separating (shared) digit identity from the (private) background is straightforward (Figure 8 in Appendix C.4); so we omit disentanglement metrics.

**MNIST-SVHN.** Moving to MNIST-SVHN, disentangling shared and private factors is more challenging due to SVHN's background clutter. Table 2 shows the *disentanglement measured directly on the latent representations*, indicating that Hölder++ notably reduces MNIST private-latent accuracy compared to Hölder+. Moreover, for Hölder++, performance on the joint $[w, z]$ and the shared $z$ differs only slightly, showing that the drop in private-latent accuracy re-

*Table 3.* Clustering performance on CUBICC using shared latent representations. We partition models according to whether $z$ follows a structured clustering prior. The best and second-best results are marked bold and underlined, respectively.

| | Image Representation (↑) | | | Caption Representation (↑) | | | Joint Representation (↑) | | |
|---|---|---|---|---|---|---|---|---|---|
| | ACC | NMI | ARI | ACC | NMI | ARI | ACC | NMI | ARI |
| MMVAE+ | $30.1 \pm 2.1$ | $16.2 \pm 2.8$ | $9.2 \pm 1.9$ | $22.9 \pm 2.6$ | $7.6 \pm 2.8$ | $3.3 \pm 2.0$ | $35.5 \pm 4.4$ | $25.4 \pm 5.2$ | $15.7 \pm 4.8$ |
| DCMEM | $34.4 \pm 28.5$ | $20.7 \pm 30.4$ | $17.4 \pm 26.9$ | $28.0 \pm 18.6$ | $13.5 \pm 19.3$ | $9.9 \pm 15.3$ | $32.8 \pm 26.5$ | $21.1 \pm 31.1$ | $17.4 \pm 27.0$ |
| Hölder+ | $28.8 \pm 3.2$ | $14.0 \pm 4.3$ | $7.9 \pm 2.8$ | $20.7 \pm 1.0$ | $4.9 \pm 1.0$ | $1.7 \pm 0.6$ | $32.3 \pm 3.4$ | $20.7 \pm 3.4$ | $11.8 \pm 2.3$ |
| Hölder++ | $28.3 \pm 2.6$ | $14.3 \pm 2.4$ | $7.7 \pm 1.8$ | $19.7 \pm 1.1$ | $3.9 \pm 0.9$ | $1.2 \pm 0.5$ | $31.8 \pm 2.4$ | $18.8 \pm 3.0$ | $10.9 \pm 2.3$ |
| CMVAE | $51.9 \pm 12.8$ | $42.2 \pm 12.1$ | $31.1 \pm 14.0$ | $\underline{44.6} \pm 13.8$ | $\mathbf{33.7} \pm 12.8$ | $\mathbf{23.8} \pm 13.1$ | $58.0 \pm 13.8$ | $51.3 \pm 12.4$ | $39.3 \pm 14.9$ |
| CHölder+ | $\mathbf{59.1} \pm 6.1$ | $\mathbf{48.8} \pm 3.4$ | $\mathbf{36.2} \pm 4.8$ | $\mathbf{45.3} \pm 4.2$ | $32.3 \pm 2.2$ | $\underline{21.3} \pm 2.8$ | $\underline{61.3} \pm 4.6$ | $\underline{51.7} \pm 2.8$ | $38.6 \pm 3.4$ |
| CHölder++ | $\underline{57.3} \pm 3.7$ | $\underline{46.4} \pm 2.1$ | $\underline{34.1} \pm 2.3$ | $44.0 \pm 3.4$ | $\underline{31.4} \pm 2.4$ | $20.5 \pm 2.4$ | $\mathbf{65.3} \pm 5.8$ | $\mathbf{55.1} \pm 3.5$ | $\mathbf{43.2} \pm 5.2$ |

*Table 4.* Disentanglement metrics on MNIST-SVHN. The best and second-best results are marked bold and underlined, respectively.

| | $w$ content accuracy ↓ | $z$ content stability ↑ | $z$ content accuracy ↑ |
|---|---|---|---|
| DMVAE | $0.168 \pm 0.006$ | $0.523 \pm 0.004$ | $0.579 \pm 0.005$ |
| MMVAE+ | $0.151 \pm 0.017$ | $0.712 \pm 0.044$ | $0.664 \pm 0.016$ |
| CMVAE | $0.139 \pm 0.015$ | $0.803 \pm 0.053$ | $0.665 \pm 0.022$ |
| DCMEM | $\mathbf{0.102} \pm 0.001$ | $\mathbf{0.853} \pm 0.006$ | $\mathbf{0.883} \pm 0.005$ |
| Hölder+ | $0.121 \pm 0.005$ | $0.800 \pm 0.008$ | $0.838 \pm 0.006$ |
| Hölder++ | $\underline{0.119} \pm 0.001$ | $\underline{0.827} \pm 0.010$ | $\underline{0.857} \pm 0.007$ |

*Table 5.* Disentanglement metrics on CUBICC. The best and second-best results are marked bold and underlined, respectively.

| Method | $w$ content accuracy ↓ | $z$ content stability ↑ | $z$ content accuracy ↑ |
|---|---|---|---|
| MMVAE+ | $0.138 \pm 0.013$ | $0.846 \pm 0.166$ | $0.624 \pm 0.049$ |
| DCMEM | $0.321 \pm 0.081$ | $0.207 \pm 0.110$ | $0.204 \pm 0.123$ |
| Hölder+ | $0.195 \pm 0.048$ | $0.442 \pm 0.087$ | $0.443 \pm 0.051$ |
| Hölder++ | $\underline{0.133} \pm 0.004$ | $0.870 \pm 0.097$ | $0.619 \pm 0.021$ |
| CMVAE | $0.136 \pm 0.011$ | $\underline{0.893} \pm 0.145$ | $\underline{0.641} \pm 0.060$ |
| CHölder+ | $0.142 \pm 0.006$ | $0.587 \pm 0.045$ | $0.535 \pm 0.030$ |
| CHölder++ | $\mathbf{0.132} \pm 0.003$ | $\mathbf{0.914} \pm 0.066$ | $\mathbf{0.642} \pm 0.014$ |

flects reduced leakage into $w$ rather than a degradation of the $z$ representation. Our models also outperform DCMEM on the SVHN modality, which is more complex than MNIST.

The *disentanglement metrics measured on the generated images* are reported in Table 4. Here, Hölder++ improves over Hölder+ by lowering $w$ content accuracy and increasing $z$ content accuracy and stability, yielding more robust generations under changes in private factors. While DCMEM achieves the best disentanglement, consistent with its high coherence, it is limited to bimodal settings; in contrast, our approach scales beyond two modalities and remains competitive in generative quality and coherence. Qualitative results in Appendix C.5, Figure 9, further show that, for cluttered SVHN inputs containing multiple digits, Hölder++ is more stable than DCMEM under variations in the private latent $w$, while maintaining accurate and high-quality generations.

**CUBICC.** Table 5 reports disentanglement metrics for

cross-modal generation with images as the target modality. Across all three metrics, hierarchical inference yields consistent gains (Hölder++ over Hölder+ and CHölder++ over CHölder+), supporting our modeling assumptions. Notably, CHölder++ achieves the strongest disentanglement with low variance, outperforming the next-best CMVAE, which exhibits much higher variance. Table 9 in Appendix C.6 reports the disentanglement on the latent representations.

**Take-away.** These results confirm that *hierarchical inference effectively prevents class information from leaking into the modality-specific representations*, especially on complex datasets such as CUBICC, while preserving generative performance and downstream-task results (see Section 4.3). This behavior is consistent with our inference design, which conditions $w$ on $z$ to capture posterior dependencies without introducing them into the generative model.

### 4.3. Downstream clustering task

**CUBICC.** We apply (bird) clustering to the latent representations on the CUBICC dataset (Palumbo et al., 2024). Table 3 shows that assuming a mixture model on the shared latent space improves the results. CHölder+ and CHölder++ rank first and second across all clustering metrics across modalities. While CMVAE is comparable to ours on the caption representation, it exhibits higher variance, similar to DCMEM, indicating sensitivity to training initialization. Refer to Figure 5 in Appendix C.6 for the plots for 10 independent runs of every model. We run paired Hotelling's $T^2$ tests (Hotelling, 1931) across 10 seeds to compare CMVAE with CHölder+ and CHölder++. Our results show that, while our improvement on the caption representation over CMVAE is not statistically significant, CHölder+ and CHölder++ provide statistically ($p < 0.05$) and marginally ($p \approx 0.05$) significant improvements, respectively, in the image representation. Remarkably, both Hölder-based models significantly outperform CMVAE in the joint representation. Finally, under the same standard-prior setting, statistical tests show that we cannot conclude a significant difference in clustering performance between MMVAE+ and our Hölder+ (see Appendix C.6 for details).

*Table 6.* Effect of hierarchical inference on CUBICC. The best and second-best results are marked bold and underlined, respectively.

| Method | $w$ content accuracy ↓ | $z$ content stability ↑ | $z$ content accuracy ↑ |
|---|---|---|---|
| MMVAE+ | $0.138 \pm 0.013$ | $0.846 \pm 0.166$ | $0.624 \pm 0.049$ |
| MMVAE++ | $0.133 \pm 0.002$ | $\mathbf{0.960} \pm \mathbf{0.008}$ | $0.640 \pm 0.010$ |
| Hölder+ | $0.195 \pm 0.048$ | $0.442 \pm 0.087$ | $0.443 \pm 0.051$ |
| Hölder++ | $0.133 \pm 0.004$ | $0.870 \pm 0.097$ | $0.619 \pm 0.021$ |
| CMVAE | $0.136 \pm 0.011$ | $0.893 \pm 0.145$ | $0.641 \pm 0.060$ |
| CMVAE++ | $\mathbf{0.130} \pm \mathbf{0.003}$ | $\underline{0.959} \pm 0.008$ | $\mathbf{0.651} \pm \mathbf{0.011}$ |
| CHölder+ | $0.142 \pm 0.006$ | $0.587 \pm 0.045$ | $0.535 \pm 0.030$ |
| CHölder++ | $\underline{0.132} \pm 0.003$ | $0.914 \pm 0.066$ | $\underline{0.642} \pm 0.014$ |

### 4.4. Effect of hierarchical inference across models

**CUBICC.** We investigate the effect of hierarchical inference across models by applying it to MMVAE+ and CM-VAE, yielding MMVAE++ and CMVAE++. Table 6 shows consistent improvements across the three disentanglement metrics for all backbones under hierarchical inference. The gains are larger for Hölder+ and CHölder+ because short-cut prevention for pairwise components is activated in the $M > 2$ setting, as in Eq. (3). Overall, MMVAE++, CM-VAE++, and CHölder++ achieve comparable disentanglement, while CHölder++ remains best in clustering (see Table 10 in Appendix C.6). These results further confirm that: (i) Hölder-based VAEs yield better trade-offs across generative performance metrics; and (ii) hierarchical inference is a robust and effective design choice for learning disentangled private-shared representations in multimodal VAEs.

## 5. Conclusion

Generative coherence is a fundamental goal of multimodal generative models. Achieving coherent samples requires learning joint representations that capture the semantics shared across modalities. This is orthogonal to generative capacity and is transversal across modeling frameworks. In this work, we focus on multimodal VAEs as a principled and scalable framework for integrating data modalities. We propose a novel multimodal VAE that improves the SOTA quality–coherence trade-off through symmetric Hölder pooling (i.e., $\alpha = 0.5$) and two key architectural choices, enabling the learning of structured latent representations.

**Summary.** First, we present the first exact implementation of symmetric Hölder pooling, whose pairwise components implicitly capture soft cross-modality dependencies by increasing regions of mutual support, unlike PoE and MoE, which assume modality independence. Second, we extend the Hölder VAE with explicit shared and modality-specific subspaces, further improving the quality-coherence trade-off, increasing sample diversity, and outperforming strong baselines such as MMVAE+. Finally, we introduce a hierar-

chical variational posterior that reduces shared-information leakage into modality-specific representations, thereby enhancing private-shared disentanglement and yielding useful representations for downstream tasks.

**Future work.** While our models obtain the best trade-offs among multimodal VAEs, they do not yet match SOTA image generators like Denoising Diffusion Probabilistic Models (DDPMs). Existing approaches close this gap via post-processing, e.g., using a DDPM to denoise VAE-generated images (Palumbo et al., 2024; Zhang et al., 2026). As future work, we will investigate how to improve generative performance within the VAE framework via latent diffusion (Wesego & Rooshenas, 2024a; Bounoua et al., 2024) or flexible priors (Wesego & Rooshenas, 2024b; Yuan et al., 2024; Senellart & Allassonnière, 2025; Oubari et al., 2026). Also, following Xiao & Bamler (2023), we will explore separate $\beta$ values for $z$ and $w$ to regularize the shared and private latent spaces and control how information is allocated between them. Finally, future work includes a theoretical analysis of our results, showing that Hölder++ can isolate shared (content) from modality-specific (style) factors.

The PyTorch implementation for our paper is available at https://github.com/vothuckkhanhhuyen/holderplusplus.

## Acknowledgements

We thank the anonymous reviewers at ICML 2026 for their helpful and constructive comments, which improved the clarity of the paper. This work has been supported by the project *"Society-Aware Machine Learning: The paradigm shift demanded by society to trust machine learning"*, funded by the European Union and led by Isabel Valera (ERC-2021-STG, SAML, 101040177). Huyen Vo acknowledges her membership in the CS@Max Planck doctoral program. Additionally, Isabel Valera and María Martínez-García acknowledge the stimulating research environment of the GRK 2853/1 *"Neuroexplicit Models of Language, Vision, and Action"*, funded by the Deutsche Forschungsgemeinschaft (DFG, German Research Foundation) under project number 471607914. Views and opinions expressed are, however, those of the author(s) only and do not necessarily reflect those of the aforementioned funding agencies. Neither of the aforementioned parties can be held responsible for them.

## Impact Statement

This paper advances multimodal VAEs by improving the quality-coherence trade-off and by promoting disentanglement between shared (content) and modality-specific (style) latent factors. These advances may enhance the interpretability and controllability of multimodal representations. Improved multimodal generative models could benefit applications that require consistent cross-modal reasoning, such

as medical image-report modeling. Nevertheless, in high-stakes domains such as healthcare, we advise practitioners to exercise caution when interpreting the inferred latent spaces, particularly with respect to causal claims, as the proposed models capture statistical dependencies between modalities rather than causal relationships. Moreover, when trained on sensitive data, practitioners should assess the model's privacy and robustness with respect to potential adversarial attacks aimed at extracting sensitive information. Beyond these considerations, we do not foresee negative societal consequences that are unique to this work, beyond those generally associated with the development and deployment of machine learning models.

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

# Hölder++: Improving the Quality-Coherence Trade-off in Multimodal VAEs
# Supplementary Material

---

**Table of Contents**

The supplementary material is organized as follows. Section A provides detailed derivations of the Hölder, Hölder+, and Hölder++ objectives, and includes proofs that each objective corresponds to a valid ELBO. Section B provides additional experimental details, including descriptions of the datasets, evaluation criteria, and training setup (architectures and hyperparameters). Section C contains further empirical results, including runtime analysis, clustering consistency across models and random seeds, effect of latent dimensionality, and additional qualitative and quantitative results on all datasets.

## A. Proofs

### A.1. Derivations of Hölder, Hölder+, and Hölder++

**Exact approximation of Hölder pooling.** Consider the Hölder pooling function with $\alpha = 0.5$, applied to the set of unimodal posteriors $\{q_{\phi_j}(\boldsymbol{z}|\boldsymbol{x}_j)\}_{j=1}^M$. Each posterior is a multivariate Gaussian distributions with diagonal covariance matrix, $q_{\phi_j}(\boldsymbol{z}|\boldsymbol{x}_j) = \mathcal{N}\left(\boldsymbol{z}; \boldsymbol{\mu}_j, \mathrm{diag}(\boldsymbol{\sigma}_j^2)\right)$, where $\boldsymbol{z} = (z_1, z_2, \ldots, z_D)^\top \in \mathbb{R}^D$ denotes the latent variable, $\boldsymbol{\mu}_j = (\mu_{j,1}, \mu_{j,2}, \ldots, \mu_{j,D})^\top \in \mathbb{R}^D$ the mean vector, and $\mathrm{diag}(\boldsymbol{\sigma}_j^2) = \mathrm{diag}(\sigma_{j,1}^2, \sigma_{j,2}^2, \ldots, \sigma_{j,D}^2)^\top \in \mathbb{R}^{D \times D}$ the covariance matrix. For simplicity, we denote the unimodal posterior by $q_j(\boldsymbol{z}) = q_{\phi_j}(\boldsymbol{z}|\boldsymbol{x}_j)$, and the approximate joint posterior by

$q(z) = q_\phi(z|X)$. The pooled density is then defined as

$$q(z) = c \left( \sum_{j=1}^{M} \lambda_j \sqrt{q_j(z)} \right)^2,$$

where $c = 1/ \int \left( \sum_{j=1}^{M} \lambda_j \sqrt{q_j(z)} \right)^2 dz$. Since determining the weights $\lambda_j$ in multimodal VAEs is generally nontrivial, we follow PoE and MoE and set them uniformly. We then obtain

$$q(z) = c \left( \sum_{j=1}^{M} \sqrt{q_j(z)} \right)^2 = c \left( \sum_{j=1}^{M} q_j(z) + 2 \sum_{i=1}^{M} \sum_{j>i}^{M} \sqrt{q_i(z)q_j(z)} \right).$$

As $q_i(z)$ and $q_j(z)$ are multivariate Gaussian distributions with diagonal covariance matrices, they factorize across dimensions as $q_i(z) = \prod_{d=1}^{D} q_{i,d}(z_d)$ and $q_j(z) = \prod_{d=1}^{D} q_{j,d}(z_d)$, where $q_{i,d}(z_d)$ and $q_{j,d}(z_d)$ denote one-dimensional Gaussian distributions along coordinate $z_d$. Hence, we can express the geometric mean $\sqrt{q_i(z)q_j(z)}$ between them as

$$\sqrt{q_i(z)q_j(z)} = \sqrt{\prod_{d=1}^{D} q_{i,d}(z_d)q_{j,d}(z_d)} = \prod_{d=1}^{D} \sqrt{q_{i,d}(z_d)q_{j,d}(z_d)}.$$

Leveraging the derivation from Vo & Valera (2026), we obtain $\sqrt{q_{i,d}(z_d)q_{j,d}(z_d)}$ for each dimension $d \in \{1, 2, \ldots, D\}$ as

$$\sqrt{q_{i,d}(z_d)q_{j,d}(z_d)} = S_{ij}^d q_{ij,d}^{(1/2)}(z_d) = S_{ij}^d \mathcal{N}(z_d; \mu_{ij,d}, \sigma_{ij,d}^2), \quad \text{where}$$

$$S_{ij}^d = \sqrt{\frac{2\sigma_{i,d}\sigma_{j,d}}{\sigma_{i,d}^2 + \sigma_{j,d}^2}} \exp\left(-\frac{1}{4}\frac{(\mu_{i,d} - \mu_{j,d})^2}{\sigma_{i,d}^2 + \sigma_{j,d}^2}\right), \quad \mu_{ij,d} = \frac{\mu_{i,d}\sigma_{j,d}^2 + \mu_{j,d}\sigma_{i,d}^2}{\sigma_{i,d}^2 + \sigma_{j,d}^2}, \quad \sigma_{ij,d}^2 = \frac{2\sigma_{i,d}^2\sigma_{j,d}^2}{\sigma_{i,d}^2 + \sigma_{j,d}^2}.$$

Then, $\sqrt{q_i(z)q_j(z)}$ becomes

$$\sqrt{q_i(z)q_j(z)} = S_{ij} q_{ij}^{(1/2)}(z) = S_{ij} \mathcal{N}(z; \boldsymbol{\mu}_{ij}, \boldsymbol{\sigma}_{ij}^2),$$

where $q_{ij}^{(1/2)}(z) = \mathcal{N}(z; \boldsymbol{\mu}_{ij}, \boldsymbol{\sigma}_{ij}^2), \boldsymbol{\mu}_{ij} = (\mu_{ij,1}, \mu_{ij,2}, \ldots, \mu_{ij,D})^\top, \boldsymbol{\sigma}_{ij}^2 = (\sigma_{ij,1}^2, \sigma_{ij,2}^2, \ldots, \sigma_{ij,D}^2)^\top$, and

$$S_{ij} = \prod_{d=1}^{D} S_{ij}^d = \prod_{d=1}^{D} \sqrt{\frac{2\sigma_{i,d}\sigma_{j,d}}{\sigma_{i,d}^2 + \sigma_{j,d}^2}} \exp\left(-\frac{1}{4}\frac{(\mu_{i,d} - \mu_{j,d})^2}{\sigma_{i,d}^2 + \sigma_{j,d}^2}\right).$$

Overall, we can see that the Hölder pooled density $q(z)$ is a weighted mixture of Gaussians

$$q(z) = \sum_{j=1}^{M} \pi_j q_j(z) + \sum_{i=1}^{M} \sum_{j>i}^{M} \pi_{ij} q_{ij}^{(1/2)}(z), \tag{4}$$

where $\pi_j = c, \pi_{ij} = 2cS_{ij}$, and $c = \left( M + 2 \sum_{i=1}^{M} \sum_{j>i}^{M} S_{ij} \right)^{-1}$.

**Hölder objective.** Given $M$ modalities $X := \{x_1, x_2, \ldots, x_M\}$, we consider the standard multimodal VAE generative model $p_\Theta(X, z) = p(z) \prod_{j=1}^{M} p_{\theta_j}(x_j|z)$, and optimize a lower bound of the log-evidence $p_\Theta(X)$ by maximizing the following objective

$$\mathcal{L}_{\text{VAE}}(x_{1:M}) = \mathbb{E}_{q_\Phi(z|X)} \left[ \log \frac{p_\Theta(X, z)}{q_\Phi(z|X)} \right], \tag{5}$$

As in MMVAE (Shi et al., 2019), we set $q_\Phi(z|X)$ to the uniform mixture, $q_\Phi(z \mid X) = \frac{1}{M} \sum_{j=1}^M q_{\phi_j}(z|x_j)$. Substituting into Eq. (5) and using linearity of expectation gives an equivalent decomposition into unimodal expectations, while keeping the denominator $q_\Phi(z|X)$ as the pooled posterior:

$$\mathcal{L}_{\text{MMVAE}}(\boldsymbol{x}_{1:M}) = \frac{1}{M} \sum_{j=1}^M \mathbb{E}_{q_{\phi_j}(z|x_j)} \left[ \log \frac{p_\Theta(\boldsymbol{X}, \boldsymbol{z})}{q_\Phi(\boldsymbol{z}|\boldsymbol{X})} \right].$$

We instead define $q_\Phi(z|X)$ via Hölder pooling in Eq. (4), which admits a mixture form

$$q_\Phi(\boldsymbol{z}|X) = \sum_{j=1}^M \pi_j \, q_{\phi_j}(\boldsymbol{z}|\boldsymbol{x}_j) + \sum_{i=1}^M \sum_{j>i}^M \pi_{ij} \, q_{ij}^{(1/2)}(\boldsymbol{z}|\boldsymbol{x}_i, \boldsymbol{x}_j), \tag{6}$$

with $\pi_j, \pi_{ij} \geq 0$ and $\sum_j \pi_j + \sum_{i=1}^M \sum_{j>i}^M \pi_{ij} = 1$. Substituting Eq. (6) into the ELBO 5 gives

$$\mathcal{L}_{\text{Hölder}}(\boldsymbol{x}_{1:M}) = \sum_{j=1}^M \pi_j \mathbb{E}_{q_{\phi_j}(\boldsymbol{z}|\boldsymbol{x}_j)} \left[ \log \frac{p_\Theta(\boldsymbol{X}, \boldsymbol{z})}{q_\Phi(\boldsymbol{z}|\boldsymbol{X})} \right] + \sum_{i=1}^M \sum_{j>i}^M \pi_{ij} \mathbb{E}_{q_{ij}^{(1/2)}(\boldsymbol{z}|\boldsymbol{x}_i, \boldsymbol{x}_j)} \left[ \log \frac{p_\Theta(\boldsymbol{X}, \boldsymbol{z})}{q_\Phi(\boldsymbol{z}|\boldsymbol{X})} \right], \tag{7}$$

which is a valid lower bound on the log-evidence $\log p_\Theta(\boldsymbol{X})$.

**Hölder+ objective.** We extend the shared-latent model by introducing modality-specific latent variables $\boldsymbol{W} := \{\boldsymbol{w}_1, \boldsymbol{w}_2, \ldots, \boldsymbol{w}_M\}$, where each $\boldsymbol{w}_j$ captures private factors of modality $\boldsymbol{x}_j$. The generative model is

$$p_\Theta(\boldsymbol{X}, \boldsymbol{z}, \boldsymbol{W}) = p(\boldsymbol{z}) \prod_{j=1}^M p_{\boldsymbol{\theta}_j}(\boldsymbol{x}_j|\boldsymbol{z}, \boldsymbol{w}_j) \, p(\boldsymbol{w}_j),$$

where we assume independent priors over the private latents $\{p(\boldsymbol{w}_j)\}_{j=1}^M$. For inference, we use a variational family that factorizes the joint posterior into a pooled encoder for the shared latent and unimodal encoders for the private latents

$$q_\Phi(\boldsymbol{z}, \boldsymbol{W}|\boldsymbol{X}) = q_{\Phi_{\boldsymbol{z}}}(\boldsymbol{z}|\boldsymbol{X}) \, q_{\Phi_{\boldsymbol{W}}}(\boldsymbol{W}|\boldsymbol{X}) = q_{\Phi_{\boldsymbol{z}}}(\boldsymbol{z}|\boldsymbol{X}) \prod_{j=1}^M q_{\boldsymbol{\phi}_{\boldsymbol{w}_j}}(\boldsymbol{w}_j|\boldsymbol{x}_j).$$

To avoid shortcut solutions in cross-modal generation, MMVAE+ (Palumbo et al., 2023) introduces auxiliary distributions over private latent variables for unobserved modalities when estimating cross-modal reconstruction terms. Concretely, when $\boldsymbol{z}$ is sampled from expert $j$, we draw $\boldsymbol{w}_j \sim q_{\boldsymbol{\phi}_{\boldsymbol{w}_j}}(\boldsymbol{w}_j|\boldsymbol{x}_j)$ for the observed modality and draw $\tilde{\boldsymbol{w}}_n \sim r_n(\boldsymbol{w}_n)$ for each modality $n \neq j$. The resulting objective can be written as

$$\mathcal{L}_{\text{MMVAE+}}(\boldsymbol{x}_{1:M}) = \frac{1}{M} \sum_{j=1}^M \mathbb{E}_{\substack{q_{\boldsymbol{\phi}_{\boldsymbol{z}_j}}(\boldsymbol{z}|\boldsymbol{x}_j) \\ q_{\boldsymbol{\phi}_{\boldsymbol{w}_j}}(\boldsymbol{w}_j|\boldsymbol{x}_j) \\ \{\tilde{\boldsymbol{w}}_n \sim r_n(\boldsymbol{w}_n)\}_{n \neq j}}} \log \left( \frac{p_{\boldsymbol{\theta}_j}(\boldsymbol{x}_j|\boldsymbol{z}, \boldsymbol{w}_j) p(\boldsymbol{z}) p(\boldsymbol{w}_j)}{q_{\boldsymbol{\phi}_{\boldsymbol{z}}}(\boldsymbol{z}|\boldsymbol{X}) q_{\boldsymbol{\phi}_{\boldsymbol{w}_j}}(\boldsymbol{w}_j|\boldsymbol{x}_j)} \prod_{n \neq j} p_{\boldsymbol{\theta}_n}(\boldsymbol{x}_n|\boldsymbol{z}, \tilde{\boldsymbol{w}}_n) \right), \tag{8}$$

where $\{r_n(\boldsymbol{w}_n)\}_{n=1}^M$ are auxiliary distributions over modality-specific latents.

Following the same principle, we extend Hölder pooling to the shared-private setting and use auxiliary distributions for modality-specific latents that are *not* inferred from data in a given mixture component, thereby preventing shortcut learning when estimating reconstruction terms for unobserved modalities. Let $q_\Phi(\boldsymbol{z}|\boldsymbol{X})$ denote the Hölder-pooled posterior with unimodal weights $\pi_j$ and pairwise weights $\pi_{ij}$ over components $q_{\boldsymbol{\phi}_{\boldsymbol{z}_j}}(\boldsymbol{z}|\boldsymbol{x}_j)$ and $q_{ij}^{(1/2)}(\boldsymbol{z}|\boldsymbol{x}_i, \boldsymbol{x}_j)$. The Hölder+ objective is

$$\mathcal{L}_{\text{Hölder+}}(\boldsymbol{x}_{1:M}) = \sum_{j=1}^M \pi_j \mathbb{E}_{\substack{q_{\boldsymbol{\phi}_{\boldsymbol{z}_j}}(\boldsymbol{z}|\boldsymbol{x}_j) \\ q_{\boldsymbol{\phi}_{\boldsymbol{w}_j}}(\boldsymbol{w}_j|\boldsymbol{x}_j) \\ \{\tilde{\boldsymbol{w}}_n \sim r_n(\boldsymbol{w}_n)\}_{n \neq j}}} \log \left( \frac{p_{\boldsymbol{\theta}_j}(\boldsymbol{x}_j|\boldsymbol{z}, \boldsymbol{w}_j) p(\boldsymbol{z}) p(\boldsymbol{w}_j)}{q_{\boldsymbol{\phi}_{\boldsymbol{z}}}(\boldsymbol{z}|\boldsymbol{X}) q_{\boldsymbol{\phi}_{\boldsymbol{w}_j}}(\boldsymbol{w}_j|\boldsymbol{x}_j)} \prod_{n \neq j} p_{\boldsymbol{\theta}_n}(\boldsymbol{x}_n|\boldsymbol{z}, \tilde{\boldsymbol{w}}_n) \right)$$

$$+ \sum_{i=1}^M \sum_{j>i}^M \pi_{ij} \mathbb{E}_{\substack{q_{ij}^{(1/2)}(\boldsymbol{z}|\boldsymbol{x}_i, \boldsymbol{x}_j) \\ q_{\boldsymbol{\phi}_{\boldsymbol{w}_i}}(\boldsymbol{w}_i|\boldsymbol{x}_i) \\ q_{\boldsymbol{\phi}_{\boldsymbol{w}_j}}(\boldsymbol{w}_j|\boldsymbol{x}_j) \\ \{\tilde{\boldsymbol{w}}_n \sim r_n(\boldsymbol{w}_n)\}_{n \notin \{i,j\}}}} \log \left( \frac{p_{\boldsymbol{\theta}_i}(\boldsymbol{x}_i|\boldsymbol{z}, \boldsymbol{w}_i) p_{\boldsymbol{\theta}_j}(\boldsymbol{x}_j|\boldsymbol{z}, \boldsymbol{w}_j) p(\boldsymbol{z}) p(\boldsymbol{w}_i) p(\boldsymbol{w}_j)}{q_{\boldsymbol{\phi}_{\boldsymbol{z}}}(\boldsymbol{z}|\boldsymbol{X}) q_{\boldsymbol{\phi}_{\boldsymbol{w}_i}}(\boldsymbol{w}_i|\boldsymbol{x}_i) q_{\boldsymbol{\phi}_{\boldsymbol{w}_j}}(\boldsymbol{w}_j|\boldsymbol{x}_j)} \prod_{n \notin \{i,j\}} p_{\boldsymbol{\theta}_n}(\boldsymbol{x}_n|\boldsymbol{z}, \tilde{\boldsymbol{w}}_n) \right). \tag{9}$$

**Hölder++ objective.** Since shared and modality-specific factors can be coupled in the posterior, we adopt a hierarchical variational posterior in which each private latent depends on both the observation and the shared latent, i.e., $q_{\phi_{w_j}}(w_j|x_j, z)$

$$q_\Phi(z, W|X) = q_{\Phi_z}(z|X)\, q_{\Phi_W}(W|X, z) = q_{\Phi_z}(z|X) \prod_{j=1}^{M} q_{\phi_{w_j}}(w_j|x_j, z). \tag{10}$$

From the Hölder+ objective in Eq. (9), substituting the hierarchical variational posterior in Eq. (10) into the ELBO yields the Hölder++ objective

$$
\mathcal{L}_{\text{Hölder++}}(x_{1:M}) = \sum_{j=1}^{M} \pi_j \mathbb{E}_{\substack{q_{\phi_{z_j}}(z|x_j) \\ q_{\phi_{w_j}}(w_j|x_j,z) \\ \{\tilde{w}_n \sim r_n(w_n)\}_{n\neq j}}} \log\left( \frac{p_{\theta_j}(x_j|z,w_j)p(z)p(w_j)}{q_{\phi_z}(z|X)q_{\phi_{w_j}}(w_j|x_j,z)} \prod_{n\neq j} p_{\theta_n}(x_n|z,\tilde{w}_n) \right)
$$

$$
+ \sum_{i=1}^{M}\sum_{j>i}^{M} \pi_{ij} \mathbb{E}_{\substack{q_{ij}^{(1/2)}(z|x_i,x_j) \\ q_{\phi_{w_i}}(w_i|x_i,z) \\ q_{\phi_{w_j}}(w_j|x_j,z) \\ \{\tilde{w}_n \sim r_n(w_n)\}_{n\notin\{i,j\}}}} \log\left( \frac{p_{\theta_i}(x_i|z,w_i)p_{\theta_j}(x_j|z,w_j)p(z)p(w_i)p(w_j)}{q_{\phi_z}(z|X)q_{\phi_{w_i}}(w_i|x_i,z)q_{\phi_{w_j}}(w_j|x_j,z)} \prod_{n\notin\{i,j\}} p_{\theta_n}(x_n|z,\tilde{w}_n) \right). \tag{11}
$$

where the highlighted terms indicate the changes relative to Hölder+.

## A.2. Lower-bound guarantee of the Hölder+ and Hölder++ objectives

**Lower bound guarantee of the Hölder+ objective.** We recall the objective of Hölder from Eq. (7)

$$
\mathcal{L}_{\text{Hölder}}(x_{1:M}) = \sum_{j=1}^{M} \pi_j \mathbb{E}_{q_{\phi_j}(z|x_j)} \left[ \log \frac{p(z)p_{\theta_j}(x_j|z)\prod_{n\neq j} p_{\theta_n}(x_n|z)}{q_\Phi(z|X)} \right]
$$

$$
+ \sum_{i=1}^{M}\sum_{j>i}^{M} \pi_{ij} \mathbb{E}_{q_{ij}^{(1/2)}(z|x_i,x_j)} \left[ \log \frac{p(z)p_{\theta_i}(x_i|z)p_{\theta_j}(x_j|z)\prod_{n\notin\{i,j\}} p_{\theta_n}(x_n|z)}{q_\Phi(z|X)} \right]. \tag{12}
$$

Inspired by the proof in MMVAE+ (Palumbo et al., 2023), for the first term corresponding to the unimodal component, we consider each term in the sum: when $z \sim q_{\phi_{z_j}}(z|x_j)$ is sampled from the unimodal encoder, we use this $z$ to compute the conditional likelihood of all $M$ modalities, including the self-reconstruction term $\log p_{\theta_j}(x_j|z)$ and the cross-modal reconstruction terms $\log p_{\theta_n}(x_n|z)$ for $n \neq j$. Building on this, and using the modality-specific encoder $q_{\phi_{w_j}}(w_j|x_j)$, we derive the lower bound on the likelihood as

$$
\log p_{\theta_j}(x_j|z) \geq \mathbb{E}_{q_{\phi_{w_j}}(w_j|x_j)} \left[ \log \frac{p_{\theta_j}(x_j|z,w_j)p(w_j)}{q_{\phi_{w_j}}(w_j|x_j)} \right], \quad \text{and} \tag{13}
$$

$$
\log p_{\theta_n}(x_n|z) = \log \mathbb{E}_{\tilde{w}_n \sim r_n(w_n)} p_{\theta_n}(x_n|z,\tilde{w}_n) \geq \mathbb{E}_{\tilde{w}_n \sim r_n(w_n)} \log p_{\theta_n}(x_n|z,\tilde{w}_n), \tag{14}
$$

where $r_n(w_n)$ is an auxiliary prior distribution specific to each target modality $n \in \{1, 2, \ldots, M\}$, and the second step follows from Jensen's inequality.

The second term in Eq. (12) corresponding to a pairwise component $(i, j)$ is analogously: when $z \sim q_{ij}^{(1/2)}(z|x_i, x_j)$, we bound the reconstruction terms for modalities $i$ and $j$ using their modality-specific posteriors $q_{\phi_{w_i}}(w_i|x_i)$ and $q_{\phi_{w_j}}(w_j|x_j)$ as in Eq. (13), while for each modality $n \notin \{i, j\}$ we use $\tilde{w}_n \sim r_n(w_n)$ and apply Eq. (14). Plugging the expressions in

Eqs. (13) and (14) into Eq. (12), we obtain the Hölder+ objective as

$$
\mathcal{L}_{\text{Hölder+}}(\boldsymbol{x}_{1:M}) = \sum_{j=1}^{M} \pi_j \mathbb{E}_{\substack{q_{\boldsymbol{\phi}_{\boldsymbol{z}_j}}(\boldsymbol{z}|\boldsymbol{x}_j) \\ q_{\boldsymbol{\phi}_{\boldsymbol{w}_j}}(\boldsymbol{w}_j|\boldsymbol{x}_j) \\ \{\tilde{\boldsymbol{w}}_n \sim r_n(\boldsymbol{w}_n)\}_{n \neq j}}} \log \left( \frac{p_{\boldsymbol{\theta}_j}(\boldsymbol{x}_j|\boldsymbol{z},\boldsymbol{w}_j)p(\boldsymbol{z})p(\boldsymbol{w}_j)}{q_{\boldsymbol{\phi}_{\boldsymbol{z}}}(\boldsymbol{z}|\boldsymbol{X})q_{\boldsymbol{\phi}_{\boldsymbol{w}_j}}(\boldsymbol{w}_j|\boldsymbol{x}_j)} \prod_{n \neq j} p_{\boldsymbol{\theta}_n}(\boldsymbol{x}_n|\boldsymbol{z},\tilde{\boldsymbol{w}}_n) \right)
$$

$$
+ \sum_{i=1}^{M} \sum_{j>i}^{M} \pi_{ij} \mathbb{E}_{\substack{q_{ij}^{(1/2)}(\boldsymbol{z}|\boldsymbol{x}_i,\boldsymbol{x}_j) \\ q_{\boldsymbol{\phi}_{\boldsymbol{w}_i}}(\boldsymbol{w}_i|\boldsymbol{x}_i) \\ q_{\boldsymbol{\phi}_{\boldsymbol{w}_j}}(\boldsymbol{w}_j|\boldsymbol{x}_j) \\ \{\tilde{\boldsymbol{w}}_n \sim r_n(\boldsymbol{w}_n)\}_{n \notin \{i,j\}}}} \log \left( \frac{p_{\boldsymbol{\theta}_i}(\boldsymbol{x}_i|\boldsymbol{z},\boldsymbol{w}_i)p_{\boldsymbol{\theta}_j}(\boldsymbol{x}_j|\boldsymbol{z},\boldsymbol{w}_j)p(\boldsymbol{z})p(\boldsymbol{w}_i)p(\boldsymbol{w}_j)}{q_{\boldsymbol{\phi}_{\boldsymbol{z}}}(\boldsymbol{z}|\boldsymbol{X})q_{\boldsymbol{\phi}_{\boldsymbol{w}_i}}(\boldsymbol{w}_i|\boldsymbol{x}_i)q_{\boldsymbol{\phi}_{\boldsymbol{w}_j}}(\boldsymbol{w}_j|\boldsymbol{x}_j)} \prod_{n \notin \{i,j\}} p_{\boldsymbol{\theta}_n}(\boldsymbol{x}_n|\boldsymbol{z},\tilde{\boldsymbol{w}}_n) \right).
$$

Thus, the Hölder+ objective is a valid ELBO. □

**Lower bound guarantee of the Hölder++ objective.** Compared to Hölder+ (Eq. (9)), Hölder++ (Eq. (11)) uses a hierarchical inference model for the private latent, i.e., $q_{\boldsymbol{\phi}_{\boldsymbol{w}_j}}(\boldsymbol{w}_j|\boldsymbol{x}_j,\boldsymbol{z})$ instead of $q_{\boldsymbol{\phi}_{\boldsymbol{w}_j}}(\boldsymbol{w}_j|\boldsymbol{x}_j)$. This changes the lower bound for the self-reconstruction term

$$
\log p_{\boldsymbol{\theta}_j}(\boldsymbol{x}_j|\boldsymbol{z}) = \log \int p_{\boldsymbol{\theta}_j}(\boldsymbol{x}_j|\boldsymbol{z},\boldsymbol{w}_j)p(\boldsymbol{w}_j)d\boldsymbol{w}_j
$$

$$
= \log \int q_{\boldsymbol{\phi}_{\boldsymbol{w}_j}}(\boldsymbol{w}_j|\boldsymbol{x}_j,\boldsymbol{z}) \frac{p_{\boldsymbol{\theta}_j}(\boldsymbol{x}_j|\boldsymbol{z},\boldsymbol{w}_j)p(\boldsymbol{w}_j)}{q_{\boldsymbol{\phi}_{\boldsymbol{w}_j}}(\boldsymbol{w}_j|\boldsymbol{x}_j,\boldsymbol{z})} d\boldsymbol{w}_j
$$

$$
= \log \mathbb{E}_{q_{\boldsymbol{\phi}_{\boldsymbol{w}_j}}(\boldsymbol{w}_j|\boldsymbol{x}_j,\boldsymbol{z})} \left[ \frac{p_{\boldsymbol{\theta}_j}(\boldsymbol{x}_j|\boldsymbol{z},\boldsymbol{w}_j)p(\boldsymbol{w}_j)}{q_{\boldsymbol{\phi}_{\boldsymbol{w}_j}}(\boldsymbol{w}_j|\boldsymbol{x}_j,\boldsymbol{z})} \right]
$$

$$
\geq \mathbb{E}_{q_{\boldsymbol{\phi}_{\boldsymbol{w}_j}}(\boldsymbol{w}_j|\boldsymbol{x}_j,\boldsymbol{z})} \left[ \log \frac{p_{\boldsymbol{\theta}_j}(\boldsymbol{x}_j|\boldsymbol{z},\boldsymbol{w}_j)p(\boldsymbol{w}_j)}{q_{\boldsymbol{\phi}_{\boldsymbol{w}_j}}(\boldsymbol{w}_j|\boldsymbol{x}_j,\boldsymbol{z})} \right],
$$

where the last step follows from Jensen's inequality. Thus, the Hölder++ objective is a valid ELBO. □

### A.3. Tighter variational lower bound via IWAE and DReG estimators

**The importance weighted autoencoder (IWAE).** IWAE (Burda et al., 2015) provides a *tighter* variational lower bound than the ELBO in Eq. (5) by using a properly weighted multi-sample importance estimator, given by

$$
\mathcal{L}_{\text{IWAE}}(\boldsymbol{x}_{1:M}) = \mathbb{E}_{\boldsymbol{z}^{1:K} \sim q_{\Phi}(\boldsymbol{z}|\boldsymbol{x}_{1:M})} \left[ \log \sum_{k=1}^{K} \frac{1}{K} \frac{p_{\Theta}(\boldsymbol{X},\boldsymbol{z}^k)}{q_{\Phi}(\boldsymbol{z}^k|\boldsymbol{X})} \right], \tag{15}
$$

with $K$ is the number of samples. In multimodal VAEs, IWAE estimator is often preferred because it typically yields higher-entropy variational posteriors, which is beneficial multimodal settings where each unimodal encoder should allocate probability mass to latent regions that explain other modalities. In MMVAE (Shi et al., 2019), under a MoE joint posterior, they extend $\mathcal{L}_{\text{IWAE}}$ in Eq. (15) via stratified sampling (Robert et al., 1999) over the $M$ modalities as follows

$$
\mathcal{L}_{\text{IWAE}}^{\text{MoE}}(\boldsymbol{x}_{1:M}) = \frac{1}{M} \sum_{j=1}^{M} \mathbb{E}_{\boldsymbol{z}^{1:K} \sim q_{\phi_j}(\boldsymbol{z}|\boldsymbol{x}_j)} \left[ \log \sum_{k=1}^{K} \frac{1}{K} \frac{p_{\Theta}(\boldsymbol{X},\boldsymbol{z}^k)}{q_{\Phi}(\boldsymbol{z}^k|\boldsymbol{X})} \right],
$$

which is a valid ELBO. In our case, under a Hölder mixture with $\alpha = 0.5$ in Eq. (6), we extend $\mathcal{L}_{\text{IWAE}}$ in Eq. (15) via stratified sampling over the $M$ modalities to obtain $\mathcal{L}_{\text{IWAE}}^{\text{Hölder}}$ as follows

$$
\mathcal{L}_{\text{IWAE}}^{\text{Hölder}}(\boldsymbol{x}_{1:M}) = \sum_{j=1}^{M} \pi_j \mathbb{E}_{\boldsymbol{z}^{1:K} \sim q_{\phi_j}(\boldsymbol{z}|\boldsymbol{x}_j)} \left[ \log \sum_{k=1}^{K} \frac{1}{K} \frac{p_{\Theta}(\boldsymbol{X},\boldsymbol{z}^k)}{q_{\Phi}(\boldsymbol{z}^k|\boldsymbol{X})} \right]
$$

$$
+ \sum_{i=1}^{M} \sum_{j>i}^{M} \pi_{ij} \mathbb{E}_{\boldsymbol{z}^{1:K} \sim q_{ij}^{(1/2)}(\boldsymbol{z}|\boldsymbol{x}_i,\boldsymbol{x}_j)} \left[ \log \sum_{k=1}^{K} \frac{1}{K} \frac{p_{\Theta}(\boldsymbol{X},\boldsymbol{z}^k)}{q_{\Phi}(\boldsymbol{z}^k|\boldsymbol{X})} \right],
$$

which remains a valid lower bound and is tighter than the ELBO.

**The doubly reparametrised gradient estimator (DReG).** As mentioned by Roeder et al. (2017); Rainforth et al. (2018), the standard IWAE gradient estimator can exhibit undesirably high variance. Tucker et al. (2019) address this by re-applying the reparameterization trick to obtain the doubly reparameterized gradient (DReG) estimator. Following prior multimodal VAE work that optimizes multi-sample objectives with DReG (Shi et al., 2019; Palumbo et al., 2023; 2024), we adopt the same estimator in our experiments for Hölder-based models.

# B. Experimental details

## B.1. Datasets

**PolyMNIST.** Sutter et al. (2021) introduced the dataset as an MNIST extension with diverse backgrounds. Each sample overlays an MNIST digit on a randomly selected $28 \times 28$ crop from each of five background images, yielding a five-modality benchmark. The digit label is the shared (semantic) factor, while background content and handwriting style are modality-specific. The dataset contains $60\,000$ training and $10\,000$ test images.

**MNIST-SVHN.** Shi et al. (2019) constructed a bimodal dataset by pairing MNIST and SVHN such that the two modalities in each pair depict the same digit class, aiming to separate conceptual complexity (digit) from perceptual complexity (e.g., color, style, size). To increase cross-domain correspondences, each instance in either dataset is randomly paired with 20 instances of the same class from the other dataset. The dataset contains $56\,068$ training and $10\,000$ test images.

**CUBICC.** Palumbo et al. (2024) introduced CUBICC, a variant of CUB Image-Captions in which each datapoint is a paired of bird image and caption. To obtain a realistic multimodal clustering benchmark, they merge fine-grained bird sub-species into coarser species-level classes, increasing intra-class variability while preserving shared semantic structure across modalities. The benchmark contains $13\,131$ image-caption pairs, split into $11\,834$ train, $638$ validation, and $659$ test samples, covering 22 sub-species grouped into 8 species (Blackbird, Gull, Jay, Oriole, Tanager, Tern, Warbler, Wren).

**CelebAMask-HQ.** Wesego & Rooshenas (2024b) introduced CelebAMask-HQ (Lee et al., 2020) in the context of multimodal VAEs, where images, masks, and attributes constitute the three modalities. All face-part masks are combined into a single black-and-white image, excluding the skin mask. Out of the 40 available attributes, 18 are selected. The dataset contains $30\,000$ samples, split into $24\,183$ training, $2\,993$ validation, and $2\,824$ test.

## B.2. Evaluation criteria

**Generative coherence.** To evaluate how well our model imputes missing modalities, we use classifier-based coherence. We train a classifier for each modality on its training samples. For unconditional coherence, we generate full multimodal samples from the prior and compute the fraction with consistent predicted labels across modalities. For conditional coherence, we condition on every non-empty subset (excluding the target modality), generate the target, report the fraction whose predicted label matches the ground truth. We report coherence averaged over all subsets and target modalities.

**Generative quality.** We use the Fréchet Inception Distance (FID) (Heusel et al., 2017) to quantify sample quality. The metric extracts features with a pretrained Inception network and measures the distance between the real and generated feature distributions. Lower FID implies higher visual realism and closer match to the data distribution.

**Disentanglement metrics.** To assess disentanglement between the shared and private subspaces, we evaluate both (i) the inferred latent representations and (ii) the generated images. For (i), we follow prior work and train a linear classifier on the latent codes, reporting classification accuracy. We expect high accuracy for the shared latent $z$ and low accuracy for the private latent $w$, since $w$ should not encode class information. For (ii), we introduce **three new generation-based metrics**: $z$ content stability ($\uparrow$), $z$ content accuracy ($\uparrow$), and $w$ content accuracy ($\downarrow$). To compute the first two, we fix $z$, sample multiple $w \sim p(w)$, decode, and classify the outputs. $z$ content stability measures *pairwise label agreement* across generated samples, while $z$ content accuracy measures correctness with respect to the ground-truth label. High values indicate that content is encoded primarily in $z$: fixing $z$ yields outputs with correct and invariant content despite changes in $w$. For $w$ content accuracy, we fix $w$, sample multiple $z \sim p(z)$, decode, and compute classification accuracy; lower values indicate that $w$ carries little content information. All three metrics are averaged over self- and cross-generation. *Note that for CUBICC we evaluate generation only when the target is an image (image → image and text → image), whereas for MNIST-SVHN we evaluate generation for both modalities as targets.*

**Downstream clustering metrics.** We evaluate clustering performance using three standard metrics: clustering accuracy (ACC), normalized mutual information (NMI), and the adjusted rand index (ARI). ACC reports the best label-matching accuracy after permuting cluster IDs via the Hungarian algorithm. NMI measures the shared information between predicted clusters and ground-truth labels, and ARI quantifies partition similarity based on pairwise assignments, corrected for chance. Higher values indicate better clustering for all three metrics.

### B.3. Experimental setups

We train all models with the reparameterization trick (Kingma & Welling, 2013) and optimize using Adam (Kingma & Ba, 2015) on NVIDIA A100-PCIE-80GB GPUs. Following prior work, we weight the KL term in the ELBO by a coefficient $\beta$ (Higgins et al., 2017), i.e., $\beta\mathrm{KL}(q_\Phi(z|X)\|p(z))$, and select $\beta$ via cross-validation over $\{1.0, 2.5, 5.0, 10.0\}$ for PolyMNIST and $\{1.0, 2.5, 5.0\}$ for MNIST-SVHN, CUBICC, and CelebAMask-HQ. For DCMEM, we choose the method-specific parameter $\alpha$ over $\{0.1, 0.5, 1.0\}$. The best value for each dataset is reported in Table 7, which also includes the optimal $\alpha$ for DCMEM. Across datasets, we use an isotropic Gaussian prior and model unimodal posteriors as diagonal-covariance Gaussians. We weight modality likelihood terms by relative dimensionality: the dominant modality is set to $1.0$ and the remaining modalities are scaled proportionally to their data dimensions (Shi et al., 2019; Sutter et al., 2021; Javaloy et al., 2022). We report mean $\pm$ standard deviation over 3 seeds, except over 10 seeds for CUBICC. For baselines, we use the original implementations: MMVAE+, CMVAE, and open source MultiVAE (Senellart & Allassonnière, 2025). *Note that in figures and tables reporting a single value per model, we select the value at the best-performing $\beta$ ( and $\alpha$ for DCMEM). In contrast, the quality-coherence trade-off figures plot results for all $\beta$ and $\alpha$ values listed above. The only exception is Figure 5, which reports multiple seeds at the best hyperparameters.*

*Table 7.* Optimal $\beta$ for all models (except DCMEM) and optimal $\alpha$ for DCMEM on all our datasets.

|  | PolyMNIST | MNIST-SVHN | CUBICC | CelebAMask-HQ |
|---|---|---|---|---|
| DMVAE |  | 1.0 |  |  |
| DCMEM |  | 1.0 | 1.0 |  |
| MMVAE+ | 2.5 | 2.5 | 1.0 | 1.0 |
| CMVAE | 2.5 | 2.5 | 1.0 | 1.0 |
| Hölder+ | 5.0 | 5.0 | 1.0 | 1.0 |
| CHölder+ |  |  | 1.0 |  |
| MMVAE++ |  |  | 1.0 |  |
| CMVAE++ |  |  | 1.0 |  |
| Hölder++ | 5.0 | 5.0 | 1.0 | 1.0 |
| CHölder++ |  |  | 1.0 |  |

**PolyMNIST.** The encoder and decoder architectures follow Palumbo et al. (2023), with ResNet backbones for all image modalities. We model each modality with a Laplace likelihood and use diagonal Gaussian priors and posteriors; for MMVAE, MMVAE+, and CMVAE we adopt Laplace priors and posteriors, consistent with their original setups. All single-shared-latent baselines are trained for 500 epochs with latent dimensionality 512, and batch size 256. For MMVAE, we set the latent dimensionality to 160. For MMVAE+, CMVAE, Hölder+, and Hölder++, we use shared and modality-specific subspaces of 32 dimensions each. We train MMVAE, MMVAE+, and CMVAE with $K = 1$ for 150, 150, 250 epochs, respectively, and batch size 256, whereas Hölder+ and Hölder++ are trained with a multi-sample objective with $K = 10$ for 50 epochs and batch size 32. For CMVAE, we set the number of clusters to 40. All models are trained with learning rate $5e^{-4}$.

**MNIST-SVHN.** The encoder and decoder architectures follow Gao et al. (2026), using ResNet backbones for all image modalities. We model each modality with a Laplace likelihood and use diagonal Gaussian priors and posteriors by default; for MMVAE, MMVAE+, and CMVAE, we instead adopt Laplace priors and posteriors. All single-shared-latent baselines are trained for 100 epochs with latent dimensionality 20. For DMVAE, MMVAE+, CMVAE, Hölder+, and Hölder++, we use shared and modality-specific latent subspaces of 10 and 4 dimensions, respectively. We train MMVAE+ and CMVAE with $K = 1$ for 30 epochs. MMVAE, Hölder+, and Hölder++ are trained with a multi-sample objective with $K = 10$ for 10 epochs; DMVAE, which is not mixture-based, is also trained for 10 epochs. For CMVAE, we set the number of clusters to 40. For DCMEM, we follow the original setup with 32-dimensional shared and private subspaces and batch size 64. Finally, all models are trained with batch size 100 and learning rate $5e^{-4}$. *Note that, for this dataset, we do not rescale likelihood terms by modality dimensionality; instead, we set the likelihood weights to $50.0$ for MNIST and $1.0$ for SVHN.*

**CUBICC.** The encoder and decoder architectures follow Palumbo et al. (2024), using a ResNet backbone for images and a convolutional network for text. We model each modality with a Laplace likelihood and use diagonal Gaussian priors and posteriors by default. All models, except DCMEM, are trained with a multi-sample objective with $K = 10$ for 300 epochs and batch size 32; DCMEM is trained for 200 epochs with batch size 64. All models use a learning rate of $1e^{-4}$ and separate shared and modality-specific latent subspaces of 64 and 32 dimensions, respectively. Moreover, for models that use a clustering prior on $z$, we set the number of clusters to a predefined value of 35.

**CelebAMask-HQ.** The encoder and decoder architectures follow Wesego & Rooshenas (2024b), using a ResNet backbone for images and segmentation masks, and an MLP for binary attributes. We model image and mask modalities with a Laplace likelihood and attributes with a Bernoulli likelihood. MMVAE+ and CMVAE use Laplace priors and posteriors, while Hölder+ and Hölder++ use diagonal Gaussian priors and posteriors. All models are trained with a multi-sample objective with $K = 1$ for 100 epochs, batch size 128, and a learning rate of $5e^{-4}$, with separate shared and modality-specific latent subspaces of 128 dimensions each. Moreover, for CMVAE, we set the number of clusters to 40.

## C. Additional results

### C.1. Additional results: Computational runtime analysis across models

In Table 8, we report the average per-batch training time on three datasets, MNIST-SVHN ($M = 2$), CelebAMask-HQ ($M = 3$), and PolyMNIST ($M = 5$), where $M$ is the number of modalities and $K$ the number of samples for ELBO estimation. The results show that the runtime gap increases with the number of modalities: for $M = 2$, the difference is relatively small, while for $M = 3$ and $M = 5$ the gap is around $1.8\times$ when comparing Hölder vs. MMVAE and Hölder+ vs. MMVAE+. Hölder++ has nearly the same batch training time as Hölder+, indicating that *the hierarchical inference does not introduce additional cost*. Thus, *we do not observe a quadratic increase in training time as $M$ grows in our benchmarks*; empirically, this suggests that the pairwise terms are not the main source of complexity, and can be masked by other factors such as latent dimensionality and model architecture. We are not currently aware of an alternative formulation or estimator that would make this computation more efficient, and we leave this for future work.

*Table 8.* Training batch time (s) across datasets. We group models by latent structure: shared-only vs. shared-private.

| Model | MNIST-SVHN ($M = 2$) | CelebAMask-HQ ($M = 3$) | PolyMNIST ($M = 5$) |
|---|---|---|---|
| MVAE | 0.0482 | 0.1442 | 0.1887 |
| MoPoE | 0.0321 | 0.0747 | 0.1216 |
| HELVAE | 0.0303 | 0.0727 | 0.0895 |
| DMVAE | 0.0537 | 0.2412 | 0.3071 |
| MMVAE ($K = 1$) | 0.0354 | 0.1717 | 0.2362 |
| Hölder ($K = 1$) | 0.0457 | 0.3252 | 0.4007 |
| MMVAE+ ($K = 1$) | 0.0501 | 0.1940 | 0.2884 |
| CMVAE ($K = 1$) | 0.0827 | 0.2456 | 0.2803 |
| Hölder+ ($K = 1$) | 0.0597 | 0.3510 | 0.4789 |
| Hölder++ ($K = 1$) | 0.0617 | 0.3518 | 0.4759 |

### C.2. Additional results: Clustering performance consistency across models

As shown in Table 3, we observe large variance in CMVAE and DCMEM. To visualize this, we plot 10 different seeds for each model under its best configuration. Figure 5 shows that CMVAE and DCMEM are widely spread in the plot, indicating sensitivity to random seeds and thus producing inconsistent results. In contrast, our Hölder-based models are more robust to initialization and regularization, provide consistent results, and achieve the best overall clustering performance.

### C.3. Additional results: Effect of latent dimensionality on generative performance

In the shared-private setting, we ablate the modality-specific latent size relative to the shared one on PolyMNIST and MNIST-SVHN in Figure 6 and 7, respectively. Hölder+ and Hölder++ are more stable and less sensitive to this choice than MMVAE+ or CMVAE, indicating that the gains stem from aggregation and inference design rather than increased capacity, as long as the latents act as an information bottleneck. In the main results, we report 10% capacity for PolyMNIST and 40% for MNIST-SVHN, matching the MMVAE+ configuration for fair comparison.

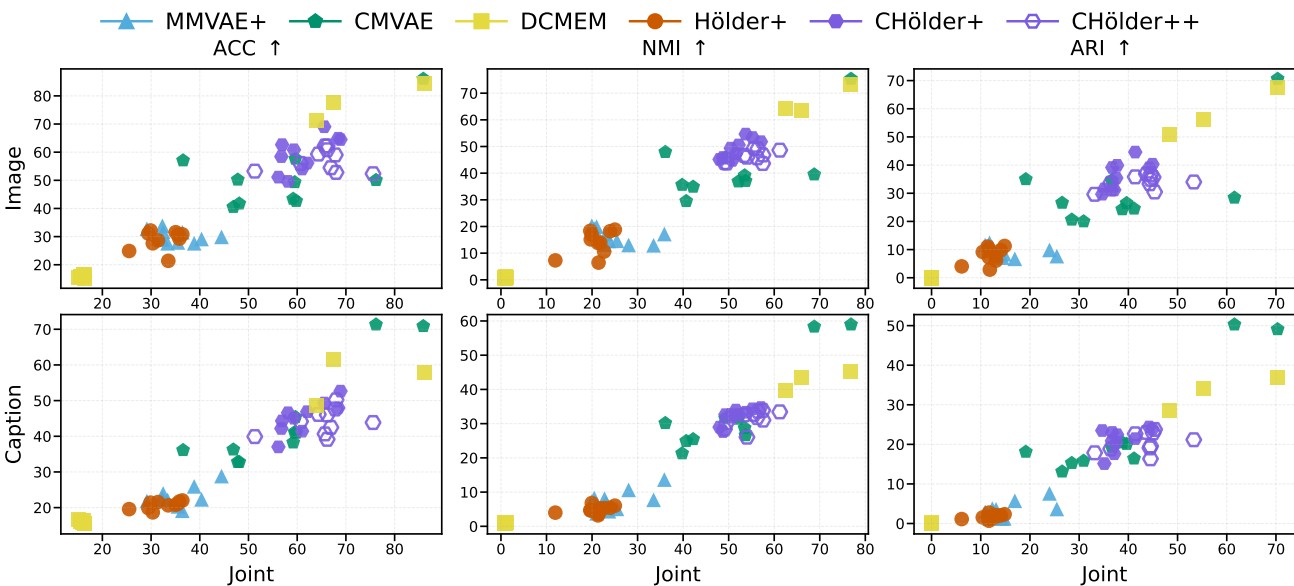

Figure 5. Clustering performance on CUBICC using latent representations, with each model evaluated at its best configuration. Per model, each point corresponds to a different seed (10 seeds total). The optimal region is the upper-right in both plots.

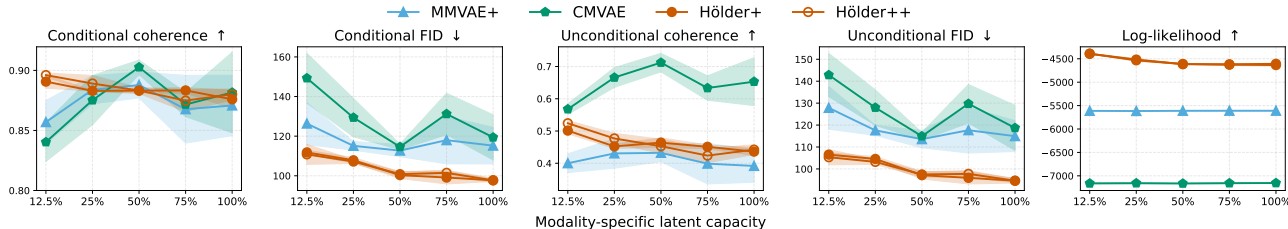

Figure 6. Generative coherence (↑), generative quality (FID ↓), and log-likelihood (↑) on PolyMNIST as a function of modality-specific latent capacity. The x-axis percentages denote the relative size of the private subspace w.r.t the shared subspace.

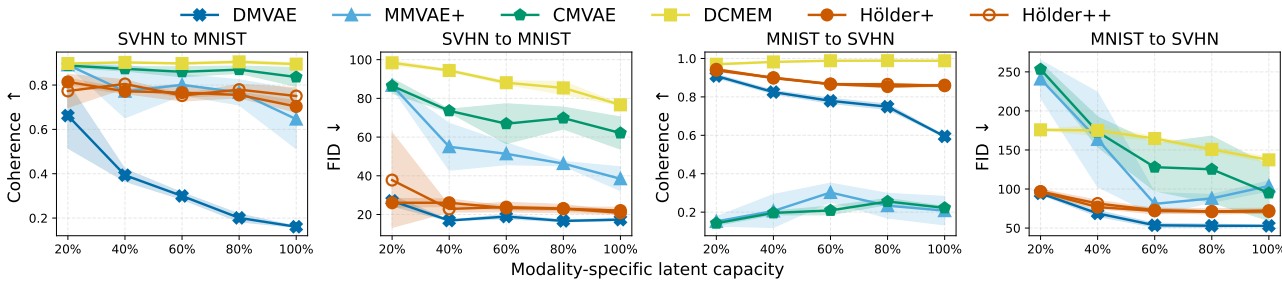

Figure 7. Generative coherence (↑) and generative quality (FID ↓) across cross-generation tasks on MNIST-SVHN as a function of modality-specific latent capacity. The x-axis percentages denote the relative size of the private subspace w.r.t. the shared subspace.

## C.4. Additional results: PolyMNIST

**Qualitative results.** Figure 8 shows cross-modal generation from the first to the third modality, with five samples obtained by varying only the modality-specific latent variables. As expected on PolyMNIST, preserving shared semantic content is straightforward: most models keep the digit identity consistent across samples. However, MMVAE+ and CMVAE still misclassify the digit for challenging cases (4 and 9, respectively). In contrast, our Hölder+ and Hölder++ preserve the correct digit information while producing samples with different styles when changing the private factors $w$.

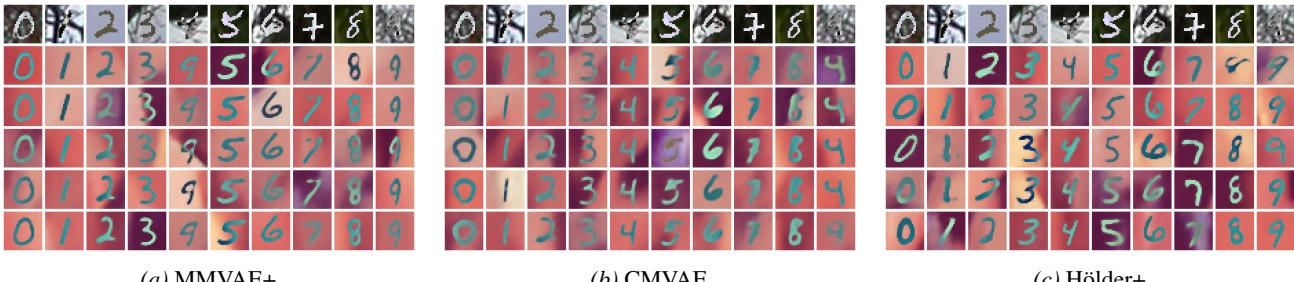

*(a)* MMVAE+        *(b)* CMVAE        *(c)* Hölder+

*Figure 8.* Five samples of the third modality conditioned on the first modality on PolyMNIST, generated by varying only the modality-specific latent variables. Each column corresponds to a ground-truth digit label from 0 to 9, so all samples within a column share the same digit information. As expected, all three models preserve the class label while changing the private factors.

### C.5. Additional results: MNIST-SVHN

**Qualitative results.** Figure 9 shows cross-modal generation from SVHN to MNIST, with five samples obtained by varying only the modality-specific latent variables. DMVAE suffers from shortcut behavior as the generated digit changes when the private factor changes. MMVAE+ and CMVAE preserve the digit, but the resulting samples are less diverse and lower quality. For DCMEM, when the SVHN input has a cluttered background (multiple digits), it struggles to predict the correct label. In contrast, our Hölder-based models consistently predict the correct label, and the improvement from Hölder+ to Hölder++ suggests that hierarchical inference better isolates shared information, yielding generations with higher quality, more diversity, and better accuracy. Similarly, Figure 10 shows cross-modal generation from MNIST to SVHN, with five samples obtained by varying the private latents $w$. MMVAE+ and CMVAE perform poorly in this setting, whereas DCMEM and our models (Hölder+ and Hölder++) preserve the digit as $w$ changes. This qualitative behavior is consistent with the coherence-quality trade-off in Figure 4 and the disentanglement metrics in Table 4.

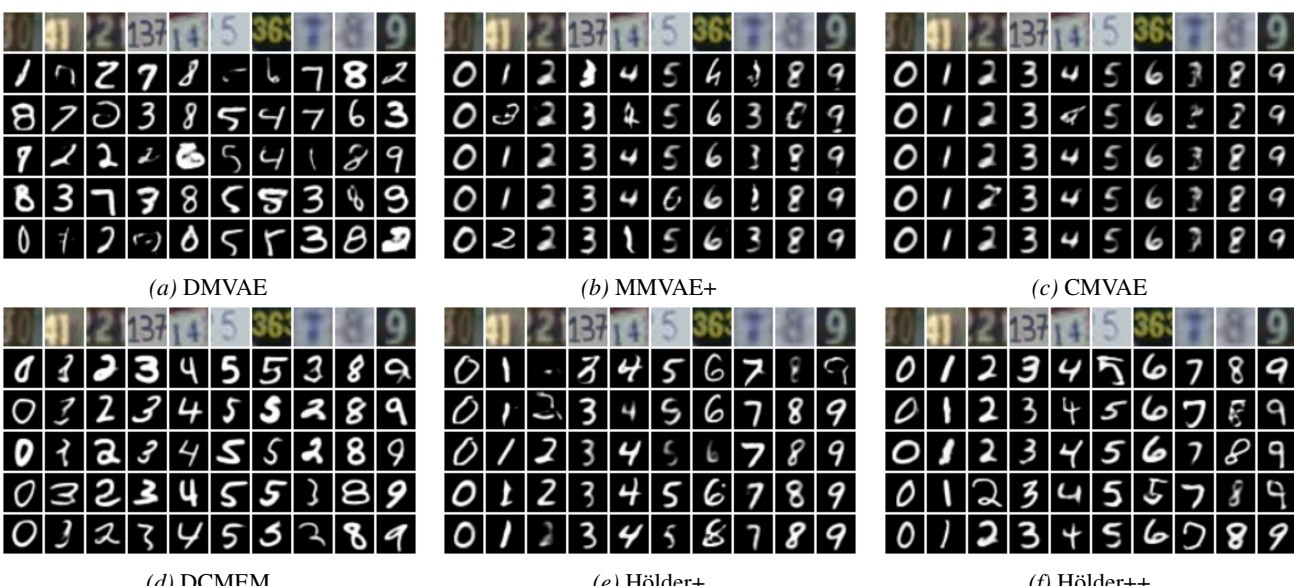

*(a)* DMVAE        *(b)* MMVAE+        *(c)* CMVAE

*(d)* DCMEM        *(e)* Hölder+        *(f)* Hölder++

*Figure 9.* Five MNIST samples generated from SVHN on MNIST-SVHN by varying only the modality-specific latent variables. Each column corresponds to a ground-truth digit label from 0 to 9, so all samples within a column share the same digit information.

### C.6. Additional results: CUBICC

**Disentanglement of shared and private subspaces.** Table 9 shows classification accuracy using latent representations from the image and caption modalities. The performance of CMVAE, CHölder+, and CHölder++ is comparable across private, shared, and joint subspaces, with each achieving the best or second-best results in this setting.

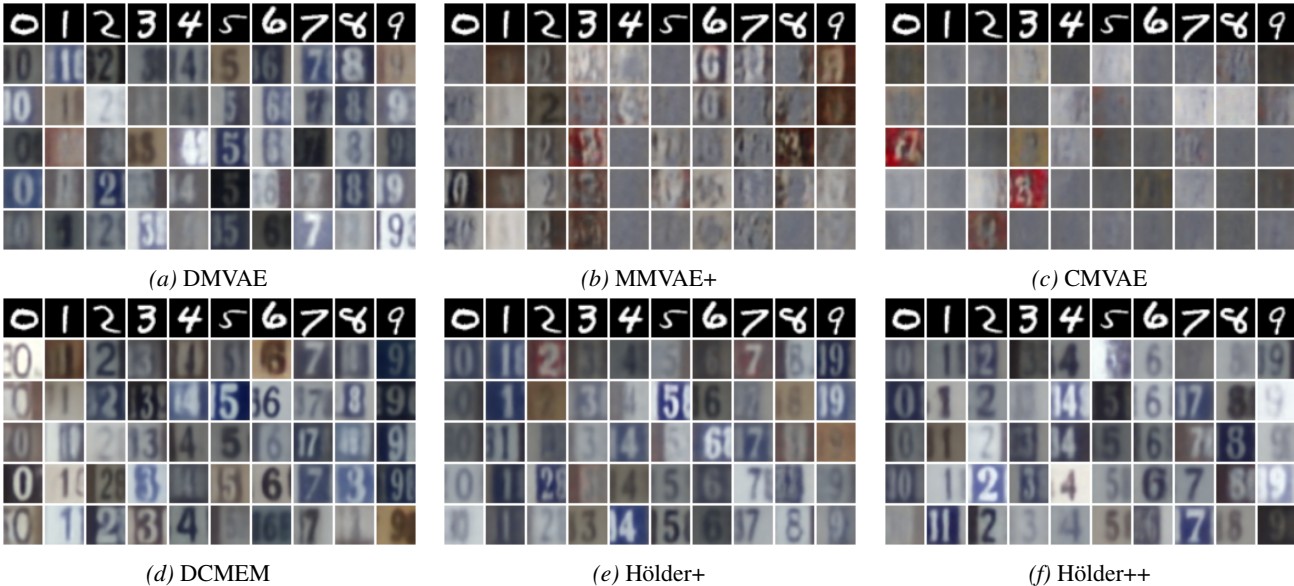

*Figure 10.* Five SVHN samples generated from MNIST on MNIST-SVHN by varying only the modality-specific latent variables. Each column corresponds to a ground-truth digit label from 0 to 9, so all samples within a column share the same digit information.

*Table 9.* Bird-species classification accuracy on the latent representations for CUBICC, evaluated on the private $w$, shared $z$, and joint $[w, z]$ subspaces. The best and second-best results are highlighted in bold and underlined, respectively.

| | Image Representation | | | Caption Representation | | |
|---|---|---|---|---|---|---|
| | Joint ($\uparrow$) | Shared ($\uparrow$) | Private ($\downarrow$) | Joint ($\uparrow$) | Shared ($\uparrow$) | Private ($\downarrow$) |
| MMVAE+ | $0.798 \pm 0.052$ | $0.796 \pm 0.045$ | $0.169 \pm 0.057$ | $\underline{0.630} \pm 0.076$ | $\underline{0.612} \pm 0.092$ | $0.231 \pm 0.050$ |
| DCMEM | $0.571 \pm 0.182$ | $0.373 \pm 0.309$ | $0.446 \pm 0.032$ | $0.470 \pm 0.114$ | $0.289 \pm 0.219$ | $0.390 \pm 0.042$ |
| Hölder+ | $\mathbf{0.827} \pm \mathbf{0.058}$ | $0.783 \pm 0.087$ | $0.311 \pm 0.059$ | $0.620 \pm 0.077$ | $0.592 \pm 0.096$ | $0.237 \pm 0.047$ |
| Hölder++ | $0.768 \pm 0.014$ | $0.758 \pm 0.017$ | $0.173 \pm 0.027$ | $0.559 \pm 0.020$ | $0.546 \pm 0.023$ | $\underline{0.207} \pm 0.014$ |
| CMVAE | $0.811 \pm 0.048$ | $\mathbf{0.810} \pm \mathbf{0.046}$ | $\mathbf{0.165} \pm \mathbf{0.039}$ | $\mathbf{0.635} \pm \mathbf{0.069}$ | $\mathbf{0.622} \pm \mathbf{0.080}$ | $0.233 \pm 0.041$ |
| CHölder+ | $\underline{0.820} \pm 0.012$ | $0.790 \pm 0.017$ | $0.234 \pm 0.029$ | $0.590 \pm 0.018$ | $0.561 \pm 0.014$ | $0.251 \pm 0.022$ |
| CHölder++ | $0.801 \pm 0.025$ | $\underline{0.800} \pm 0.023$ | $\mathbf{0.165} \pm \mathbf{0.024}$ | $0.582 \pm 0.011$ | $0.574 \pm 0.010$ | $\mathbf{0.200} \pm \mathbf{0.014}$ |

*Table 10.* Effect of hierarchical inference on clustering performance across models on CUBICC. We partition models according to whether $z$ follows a structured clustering prior. The best and second-best results are marked bold and underlined, respectively.

| | Image Representation ($\uparrow$) | | | Caption Representation ($\uparrow$) | | | Joint Representation ($\uparrow$) | | |
|---|---|---|---|---|---|---|---|---|---|
| | ACC | NMI | ARI | ACC | NMI | ARI | ACC | NMI | ARI |
| MMVAE+ | $30.1 \pm 2.1$ | $16.2 \pm 2.8$ | $9.2 \pm 1.9$ | $22.9 \pm 2.6$ | $7.6 \pm 2.8$ | $3.3 \pm 2.0$ | $35.5 \pm 4.4$ | $25.4 \pm 5.2$ | $15.7 \pm 4.8$ |
| MMVAE++ | $29.4 \pm 3.0$ | $15.1 \pm 2.4$ | $8.3 \pm 1.8$ | $20.9 \pm 2.0$ | $4.8 \pm 1.5$ | $1.7 \pm 0.9$ | $36.1 \pm 2.7$ | $24.2 \pm 2.2$ | $15.1 \pm 1.7$ |
| Hölder+ | $28.8 \pm 3.2$ | $14.0 \pm 4.3$ | $7.9 \pm 2.8$ | $20.7 \pm 1.0$ | $4.9 \pm 1.0$ | $1.7 \pm 0.6$ | $32.3 \pm 3.4$ | $20.7 \pm 3.4$ | $11.8 \pm 2.3$ |
| Hölder++ | $28.3 \pm 2.6$ | $14.3 \pm 2.4$ | $7.7 \pm 1.8$ | $19.7 \pm 1.1$ | $3.9 \pm 0.9$ | $1.2 \pm 0.5$ | $31.8 \pm 2.4$ | $18.8 \pm 3.0$ | $10.9 \pm 2.3$ |
| CMVAE | $51.9 \pm 12.8$ | $42.2 \pm 12.1$ | $31.1 \pm 14.0$ | $\underline{44.6} \pm 13.8$ | $\mathbf{33.7} \pm \mathbf{12.8}$ | $\mathbf{23.8} \pm \mathbf{13.1}$ | $58.0 \pm 13.8$ | $51.3 \pm 12.4$ | $\underline{39.3} \pm 14.9$ |
| CMVAE++ | $43.6 \pm 3.0$ | $33.7 \pm 3.8$ | $22.6 \pm 2.7$ | $37.7 \pm 4.1$ | $25.3 \pm 3.9$ | $15.9 \pm 3.3$ | $53.2 \pm 6.3$ | $45.0 \pm 4.7$ | $32.6 \pm 4.8$ |
| CHölder+ | $\mathbf{59.1} \pm \mathbf{6.1}$ | $\mathbf{48.8} \pm \mathbf{3.4}$ | $\mathbf{36.2} \pm \mathbf{4.8}$ | $\mathbf{45.3} \pm \mathbf{4.2}$ | $\underline{32.3} \pm 2.2$ | $\underline{21.3} \pm 2.8$ | $\underline{61.3} \pm 4.6$ | $\underline{51.7} \pm 2.8$ | $38.6 \pm 3.4$ |
| CHölder++ | $\underline{57.3} \pm 3.7$ | $\underline{46.4} \pm 2.1$ | $\underline{34.1} \pm 2.3$ | $44.0 \pm 3.4$ | $31.4 \pm 2.4$ | $20.5 \pm 2.4$ | $\mathbf{65.3} \pm \mathbf{5.8}$ | $\mathbf{55.1} \pm \mathbf{3.5}$ | $\mathbf{43.2} \pm \mathbf{5.2}$ |

**Effect of hierarchical inference across models.** As shown in Table 6, hierarchical inference consistently improves three disentanglement metrics across all models. Table 10 reports the corresponding downstream clustering performance in the same setting. For MMVAE+ and Hölder+, hierarchical inference does not affect clustering performance. While CMVAE is highly sensitive to random seeds under the original architecture, adding hierarchical inference (CMVAE++) substantially reduces variance across seeds, leading to more consistent and stable performance. Overall, introducing a hierarchical

posterior improves disentanglement between shared and private latent variables without hurting downstream performance, and CHölder+ and CHölder++ remain the best-performing models for clustering.

**Downstream clustering task.** From Table 10, we compare Hölder+ and Hölder++ with MMVAE+ and MMVAE++ under the same standard-prior setting by running multivariate Hotelling's $T^2$ tests for image, caption, and joint representations across 10 seeds, *using a significance level of 0.05*. For MMVAE+ vs. Hölder+, the differences are not statistically significant for image ($p = 0.503$), caption ($p = 0.192$), or joint ($p = 0.210$) representations. For MMVAE++ vs. Hölder++, while the joint representation shows a marginal difference ($p = 0.066$), the image ($p = 0.681$) and caption ($p = 0.598$) comparisons are not statistically significant. Therefore, *under the standard prior*, we cannot conclude a significant difference between these models. In contrast, *with a mixture prior* as shown in Section 4.3, our CHölder+ and CHölder++ achieve better clustering results than CMVAE, with significant gains in the joint representation.

**Qualitative results.** Figure 11 shows the captions when conditioning on images. We see that Hölder+ improves caption quality over MMVAE+, producing more reasonable text relative to the image. Hölder++ also performs well in this setting.

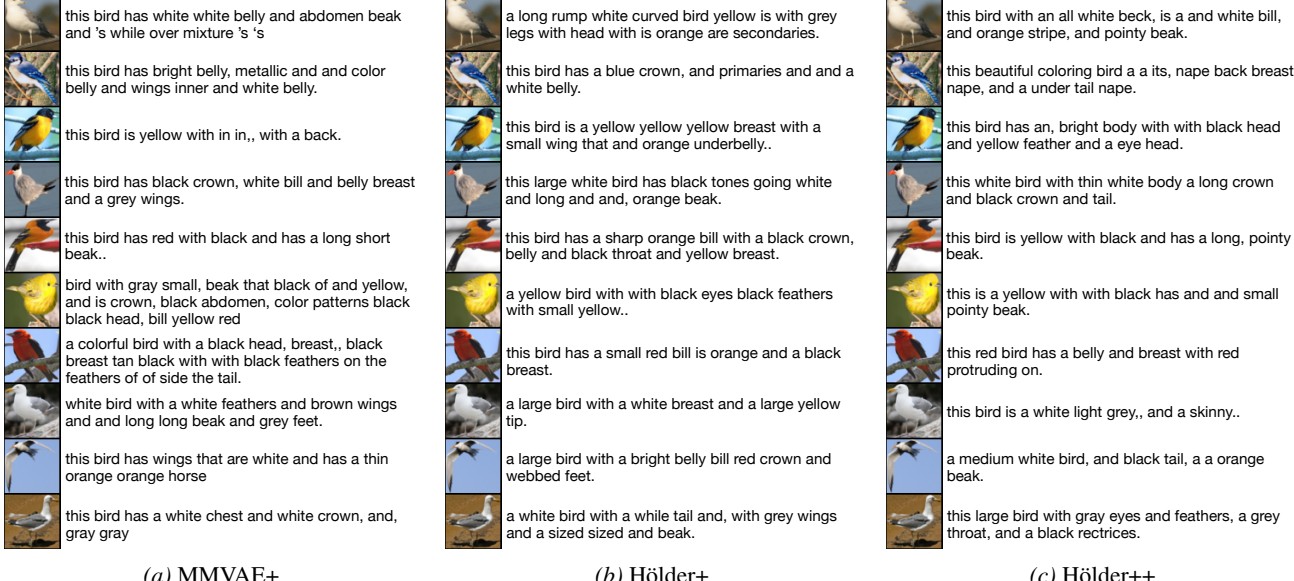

*(a)* MMVAE+      *(b)* Hölder+      *(c)* Hölder++

*Figure 11.* Qualitative results for MMVAE+, Hölder+ and Hölder++ for image-to-caption generation on CUBICC.

## C.7. Additional results: CelebAMask-HQ

**Qualitative results.** To further improve the image quality of our methods, we apply a diffusion model as a post-hoc refinement step. Figure 12 shows that feeding Hölder++ samples into a pretrained diffusion model DiffuseVAE (Pandey et al., 2024) improves image quality without altering the underlying characteristics. This confirms that our methods learn informative latent representations that can be effectively leveraged by more advanced generative models.

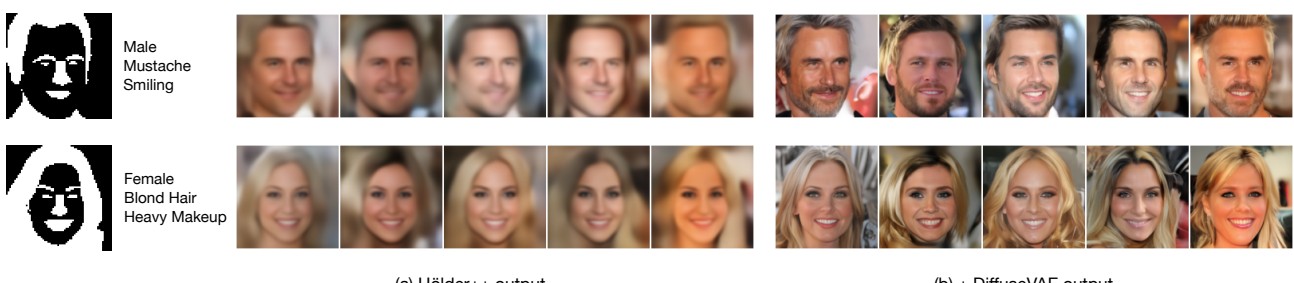

(a) Hölder++ output      (b) + DiffuseVAE output

*Figure 12.* Higher-quality image generation on CelebAMask-HQ obtained by applying DiffuseVAE as a post hoc refinement process, conditioned on the mask and attributes shown in the first two columns.

