# OpenReview forum: "Hölder++: Improving Quality-Coherence Trade-off in Multimodal VAEs"
_ICML.cc/2026/Conference — ICML 2026 regular_

### Official Review · Reviewer_c1xJ · 2026-03-09

**Soundness:** 3
**Presentation:** 3
**Significance:** 2
**Originality:** 2
**Overall Recommendation:** 4
**Confidence:** 3

**Summary:**

This paper introduces a number of new techniques for training multi-modal variational autoencoders. The authors call the collection of these techniques together: Holder++. The first contribution that the authors make is to extend previous work using Holder pooling. This technique creates a single variational posterior for the latent code by aggregating the approximate posteriors generated by separate encoders for each mode, doing so by minimizing an $\alpha$ divergence between the approximate aggregated posterior and each individual mode's posterior. Previous work approximated the pooled distribution using a Laplace approximation, while in this work the authors propose using a full mixture distribution. In their loss, the authors explicitly marginalize over the $M^2$ mixture components. The authors then apply a technique from the MMVAE+, introducing mode-specific latent variables in addition to the shared latent variable. To prevent these mode-specific variables from capturing all of the information, the authors enforce that each can only be used when the sample for mixture components that include posterior information from the correct mode. Finally to encourage further disengagement between the shared and mode-specific latents, the authors restructure the mode specific inference networks to also incorporate the shared latent as input. To evaluate their method, the authors perform experiments on two synthetic datasets based on digits and a dataset of pairs of bird images with corresponding text descriptions.

**Compliance With Llm Reviewing Policy:**

Affirmed.

**Final Justification:**

Rebuttal addressed many of my concerns, but overall significance is still somewhat small.

**Key Questions For Authors:**

- I recognize that this is an established line of work, but as I'm not entirely familiar with prior multimodal VAEs, it would be helpful to have clarification on what eventual use cases might be for multi-modal VAEs and why this style of model would be more appropriate than models that don't make the same assumptions (e.g. conditional independence given the latent variable, factorization of the approximate posterior).
- Following the discussion in weaknesses, how would you apply this to a case with many modalities where the quadratic scaling factor is impractical?
- How did you determine latent dimensionality for each model? For some of the experiments it seems that the single-shared latent baselines have higher latent dimensionality (e.g. 512 vs. 160 + 32 per mode for PolyMNIST)
- Could part of the benefit of the hierarchical encoder simply be increased flexibility? Would a hierarchical shared latent have a similar benefit?

**Limitations:**

Yes

**Strengths And Weaknesses:**

**Strengths**
- The paper is clear and well-written, particularly in its discussion of related prior work and the relationship to the proposed method.
- Based on this prior work, the methods introduced appear to be reasonably sound and somewhat well-motivated. I have no reason to doubt the claims made by the authors.
- The authors show the benefits of each one of their proposed improvements. It appears that some, such as the hierarchical posterior update are easily adapted into prior work and can provide benefits.

**Weaknesses**
- The exact holder pooling introduces a quadratic factor in computing the loss. While most of the experiments in the paper only have 2 modalities, this factor could be very significant if the number of modes increases. The authors do acknowledge and discuss this concern, but don't discuss whether it could be feasibly addressed with sampling or another estimate.
- The methodological novelty is limited, as the authors are making minor changes to prior work (e.g. enumerating mixture components rather than using a Laplace approximation, adding $z$ as an input to the mode-specific encoder) and the proposed changes don't introduce fundamentally new techniques or theory
- The significance of the work is limited as it only discusses improvements within the narrow scope of this class of multi-modal variational auto encoders. The authors don't justify why this class of models is particularly significant or useful, especially since other classes of models (diffusion, autoregressive models) that can handle multi-modal data generally have vastly better generative performance. The authors don't provide comparisons to other classes of models.
- The authors only explore 1 relatively small realistic dataset in their experiments and do not evaluate generation quality on this dataset, only disentanglement and classification metrics. The experiments show improvements over prior work, but in some cases these improvements are marginal or mixed.

---

> ### Author Rebuttal · Authors · 2026-03-31
>
> We thank the reviewer for the constructive feedback. We will revise the manuscript to address these comments by clarifying our work’s significance and originality, expanding the discussion of disentanglement via hierarchical inference, adding CelebAMask-HQ results showing improved generative quality, and including a runtime analysis showing only $\sim 1.8 \times$ overhead while pushing the Pareto front. For details, we refer the reviewer to our responses to Reviewers Lmmc and qjBk. We would appreciate your reconsideration of the score in light of these revisions and are happy to address any further feedback. Below, we address your specific concerns.
>
> ---
>
> **Significance and originality**
>
> As highlighted by Reviewers Lmmc and qjBk, the quality–coherence trade-off is an important problem, and our method is technically strong, with an exact pooled-posterior decomposition enabling tractable inference and structured latent representations. While it builds on existing components, we combine them in a principled, non-trivial way, providing the first exact implementation of Hölder pooling as an aggregation function.
>
> Beyond generation, multimodal VAEs provide a principled framework for joint multimodal analysis, learning structured representations without ad hoc alignment (e.g., ProLIP [1]), handling missing modalities and cross-modal generation, and scaling beyond image–text settings. The shared latent space is well suited for downstream analysis of high-dimensional multimodal data, including multi-omics [2], medical imaging [3], and video [4].
>
> **Improvements in performance**
>
> Multimodal VAEs face a quality–coherence trade-off: high-quality models often fail to capture shared semantics, while coherent models tend to sacrifice sample quality. We show that our method improves this trade-off. Although individual metrics are mixed, joint analysis shows that our method pushes the Pareto front (Figs. 2, 4 in the paper).
>
> **Generation quality on real-world datasets**
>
> We run additional experiments on CelebAMask-HQ, which has 3 modalities. Compared to CUBICC, its larger test set yields more stable FID estimates. As shown in the table below, Hölder-based models achieve better FID for conditional image generation. Given the character limit, we omit coherence results and will include them in the revision.
>
> To improve image quality, we apply a diffusion model as a post-hoc refinement step. Fig. 1 **in the link below** shows that feeding Hölder+ samples into a pretrained diffusion model [5] improves image quality without changing the underlying characteristics.
>
> ||Mask to Image (FID↓) |Attribute to Image (FID↓)|
> |--:|:--:|:--:|
> |MMVAE+|92.63 ± 4.67|110.15 ± 6.81|
> |Hölder+|**72.32 ± 2.11**|**87.19 ± 1.22**|
> |Hölder++|73.64 ± 6.52|90.99 ± 8.66|
>
> **Computational complexity**
>
> We refer the Reviewer to our response to Reviewer Lmmc for details. In brief, our results show that Hölder-based models improve the quality–coherence trade-off. While runtime increases moderately with the number of modalities ($\sim 1.8 \times$ overhead at $M \leq 5$), we do not observe quadratic scaling in practice.
>
> **Latent dimensionality**
>
> We match the MMVAE+ configuration for fair comparison, ensuring that the observed gains come from our design rather than model capacity. In single-shared latent models, all capacity is assigned to one latent, requiring higher dimensionality.
>
> In the shared–private setting, we ablate the modality-specific latent size relative to the shared one on PolyMNIST and MNIST-SVHN (see Figs. 2 and 3 **in the link below**). Hölder+ and Hölder++ are more stable and less sensitive to this choice than MMVAE+ or CMVAE, indicating that the gains come from aggregation and inference design rather than increased capacity, as long as the latents act as an information bottleneck. In the paper, we report 100% capacity for PolyMNIST and 40% for MNIST-SVHN.
>
> **Hierarchical latent representations**
>
> We refer the Reviewer to our response to Reviewer qjBk. In short, our variational factorization introduces a strong inductive bias that promotes disentanglement between shared and private latents, a key objective in the literature.
>
> A hierarchical shared representation, as suggested by our clustering experiments, can increase flexibility and improve performance, but would not enforce disentanglement.
>
> **Figures:** https://postimg.cc/gallery/sgXdqmj
>
> ---
>
> **References**
>
> [1] Chun et al. Probabilistic Language-Image Pre-Training, ICLR 2025.
>
> [2] Minoura et al. A mixture-of-experts deep generative model for integrated analysis of single-cell multiomics data, Cell Reports Methods 2021.
>
> [3] Dao et al. Longitudinal Alzheimer's Disease Progression Prediction with Modality Uncertainty and Optimization of Information Flow, IEEE JBHI 2024.
>
> [4] Mao et al. Multimodal Variational Auto-encoder based Audio-visual Segmentation, ICCV 2023.
>
> [5] Pandey et al. DiffuseVAE: Efficient, Controllable and High-Fidelity Generation from Low-Dimensional Latents, TMLR 2022.

---

> > ### Author Rebuttal · Reviewer_c1xJ · 2026-04-04
> >
> > Thank you to the authors for their clarifications and additional experiments. Most of my questions have been addressed, so I will raise my score, though I still see the overall significance as somewhat minor.

---

> > > ### Author Response · Authors · 2026-04-07
> > >
> > > We sincerely thank the reviewer for reconsidering the paper and for raising the score. We are glad that our response helped address the concerns.
> > >
> > > In light of the comments, we will revise the manuscript to incorporate the additional results and clarifications discussed in the rebuttal, including a discussion of the significance and originality of our work, a more detailed explanation of how hierarchical inference encourages disentanglement, and the CelebAMask-HQ generation results.
> > >
> > > We appreciate the reviewer's helpful and constructive feedback, which has helped us improve the overall quality of the paper.

---

### Official Review · Reviewer_qjBk · 2026-03-11

**Soundness:** 3
**Presentation:** 3
**Significance:** 3
**Originality:** 3
**Overall Recommendation:** 4
**Confidence:** 4

**Summary:**

This paper proposes a new multimodal VAE framework based on Hölder aggregation to improve shared latent fusion, and then extends it with private modality-specific latents and a hierarchical inference design.

**Compliance With Llm Reviewing Policy:**

Affirmed.

**Key Questions For Authors:**

For the hierarchical inference design, why is the private latent inferred from both the input and the shared latent, i.e., $q(w_j|x_j,z)$, instead of only from the shared latent, e.g., $q(w_j|z)$? if z is intented to act as an information bottleneck, it is unclear why conditioning w on x is still necessary. As stated by the author at L264-265 that "we **ensure** that wj does not capture all the information in data", I feel it would be good to clarify whether this choice is required for expressiveness, or whether a stricter hierarchical factorization would also work.

**Limitations:**

See Weaknesses and Question.

**Strengths And Weaknesses:**

A clear strength is that the paper has a strong technical core. The Hölder aggregation is not just presented as an intuition. The paper derives an exact decomposition of the pooled posterior into unimodal and pairwise terms, and this gives a tractable way to do multimodal inference. In general, I think this is good work.

I don't have major concerns about this work, but I have a slight recommendation: the Hölder pooling derivation is in good shape for the symmetric, equal-weight case, but the paper moves from the general weighted form to the equal-weight expansion a bit too quickly. The algebra is fine, but the presentation may confuse readers. In addition, for the hierarchical posterior design guarantees, the model design clearly gives a better inductive bias for separating shared and private information, but the derivation does not prove that shortcut learning is ruled out or that the private latents must only capture residual modality-specific information. It would be better for the author to further elaborate on this.

---

> ### Author Rebuttal · Authors · 2026-03-31
>
> We thank the reviewer for their constructive feedback. We are encouraged that the reviewers find (1) the paper well-written (Reviewers Lmmc, c1xJ); (2) the quality–coherence trade-off problem important and well-motivated (Reviewers Lmmc, qjBk); and (3) our method technically strong (Reviewers Lmmc, qjBk), with an exact pooled-posterior decomposition enabling tractable multimodal inference (Reviewer qjBk), more structured latents for downstream tasks (Reviewer Lmmc), and a hierarchical posterior that is easy to adapt to prior work (Reviewer c1xJ).
>
> We are revising our manuscript to incorporate all reviewers’ feedback. Key updates include:
>
> 1. Clarify the paper’s significance (a framework for multimodal generation and structured representation learning) and originality (the first exact implementation of Hölder pooling with the best quality–coherence trade-off in the current SOTA) (Reviewer c1xJ).
>
> 2. Elaborate on how hierarchical inference encourages disentanglement (Reviewer qjBk), and add CelebAMask-HQ results showing our model outperforms SOTA on generative quality (Reviewer c1xJ).
>
> 3. Add batch runtime analysis showing that, even in the worst-case setting, our method requires only about $1.8\times$ the time of MMVAE-based baselines while effectively pushing the Pareto front (Reviewer Lmmc).
>
> We would greatly appreciate your reconsideration of the score in light of these improvements and would be happy to provide further clarification if needed. Below, we address your specific concerns.
>
> ---
>
> **Uniform pooling weights**
>
> We agree that the transition from the general weighted form to the equal-weight form is too quick, and we will expand the intermediate steps in the revised version to improve readability. As shown in HELVAE, learnable pooling weights are often difficult to tune and can degrade quality and coherence on datasets such as CUB and CelebA by allowing a few modalities to dominate. Therefore, following PoE/MoE and HELVAE, we set them uniformly.
>
> **Hierarchical inference to enforce disentanglement. Why do we condition $w_j$ on both $z$ and $x$?**
>
> While we acknowledge that a formal guarantee is not included and will be left to future work, our empirical results demonstrate that the proposed factorization of the variational posterior introduces a strong inductive bias that consistently promotes disentanglement between shared and private latent variables.
>
> Importantly, the true posterior generally does not factorize over shared and private latents, even when independence is assumed in the priors, as in our setting. Our variational factorization should therefore be viewed as a deliberate modeling choice to better approximate the posterior by first inferring the shared representation $z$ (capturing shared semantics across modalities), and then the modality-specific latent $w_j$​ capturing residual information in $x_j$ that is not explained by $z$ (e.g., syntax in text or backgrounds in images).
>
> This conditioning on the observations is also related to literature in top-down hierarchical VAEs (e.g., Ladder VAE [1], BIVA [2], or NVAE [3]), where high-level latents are inferred first and lower-level latents refine the posterior by incorporating observation-specific information.
>
> This structured approximation proves effective in practice, consistently improving disentanglement and yielding gains across architectures such as Hölder++ and MMVAE++ (Hölder+ and MMVAE+ with hierarchical inference).
>
> ***Remark on the broader significance of these results***
>
> Generative coherence is a fundamental goal of multimodal generative models. Achieving coherent samples requires learning joint representations that capture semantics shared across modalities. This is orthogonal to generative capacity and is transversal across modeling frameworks.
>
> In this work, we focus on multimodal VAEs as a principled framework for integrating data modalities. We improve the quality–coherence trade-off by learning structured latent representations that can be leveraged beyond VAEs. In our response to Reviewer c1xJ, we show that combining our method with post-hoc diffusion-based refinement yields high-quality images while preserving cross-modal semantics.
>
> We therefore believe our work has broader impact, as it suggests promising directions for more principled integration with existing frameworks. This includes, for example, applying diffusion models in the shared latent space or replacing deterministic embedding models such as CLIP [4] with probabilistic joint representations, moving from conditioning-based pipelines toward unified multimodal generative modeling.
>
> ---
>
> **References**
>
> [1] Sønderby et al. Ladder Variational Autoencoders, NeurIPS 2016.
>
> [2] Maaløe et al. BIVA: A Very Deep Hierarchy of Latent Variables for Generative Modeling, NeurIPS 2019.
>
> [3] Vahdat et al. NVAE: A Deep Hierarchical Variational Autoencoder, NeurIPS 2020.
>
> [4] Radford et al. Learning transferable visual models from natural language supervision, ICML 2021.

---

### Official Review · Reviewer_Lmmc · 2026-03-13

**Soundness:** 3
**Presentation:** 3
**Significance:** 2
**Originality:** 3
**Overall Recommendation:** 5
**Confidence:** 4

**Summary:**

This paper proposes Hölder++, a novel multimodal VAE designed to improve the trade-off between generative quality and cross-modal coherence. The method combines exact Hölder pooling with a latent structure that separates shared and modality-specific information (Hölder+), and further introduces hierarchical inference to obtain more disentangled and useful representations (Hölder++).

**Compliance With Llm Reviewing Policy:**

Affirmed.

**Final Justification:**

Regarding my first point on the performance differences under the standard prior setting, the results of Hotelling’s $T^2$ tests, which show no statistically significant difference, have resolved my concern. Regarding my second point on scalability, I acknowledge the additional experimental results (M=2,3) showing that a quadratic explosion in training time is not empirically observed within the M≤5 range, which supports the authors' claims regarding computational cost in this regime. I appreciate the authors' agreement to restrict their claims to the tested settings (M≤5) and to add a Remark discussing the limitations and alternative approaches when the number of modalities scales further.
As all of my concerns have been addressed, I am raising my score to Accept, on the condition that these revisions and the additional experimental results are explicitly incorporated into the revised manuscript.

**Key Questions For Authors:**

I would like the authors to address the concerns raised in the Weaknesses section.

**Limitations:**

Yes

**Strengths And Weaknesses:**

Strengths:
- The paper is well-written and easy to read.
- The paper tackles an important and well-motivated problem in multimodal generative modeling, namely the tension between sample quality and semantic coherence across modalities, and the proposed method is technically meaningful and convincing.
- The experiments suggest that the method can produce better latent organization and more informative shared representations for downstream tasks, which makes the approach appealing beyond pure generation performance.

Weaknesses:
- The explanation of the CUBICC clustering results is currently not fully convincing. The paper attributes the weaker clustering performance of the proposed method, relative to CMVAE, to the lack of a clustering-oriented mixture prior, and uses the CHölder variant to support this interpretation. However, Table 8 shows that Hölder++ also underperforms MMVAE++ under the same standard-prior setting. Since MMVAE++ likewise does not rely on a mixture prior, the current argument based on prior choice alone is insufficient to explain the observed performance gap.
- In Appendix C.1 (Table 7), the authors acknowledge that the proposed method incurs O(M^2) computational components per batch because it explicitly models pairwise terms, but argue that the overall training time remains comparable to other methods because the model converges in fewer epochs. However, this empirical justification appears to be supported only on PolyMNIST, which has five modalities (M=5). This evidence seems insufficient to support a broader claim regarding scalability. As the number of modalities increases, the per-batch training cost will necessarily grow quadratically with M, whereas this paper does not provide either theoretical justification or empirical evidence showing that the required number of epochs would decrease enough to offset this increase in more large-scale multimodal settings. Therefore, the statement that the total training time is “comparable” appears too strong unless it is explicitly restricted to relatively small numbers of modalities.

---

> ### Author Rebuttal · Authors · 2026-03-31
>
> We thank the reviewer for their constructive feedback. We are encouraged that the reviewers find (1) the paper well-written (Reviewers Lmmc, c1xJ); (2) the quality–coherence trade-off problem important and well-motivated (Reviewers Lmmc, qjBk); and (3) our method technically strong (Reviewers Lmmc, qjBk), with an exact pooled-posterior decomposition enabling tractable multimodal inference (Reviewer qjBk), more structured latents for downstream tasks (Reviewer Lmmc), and a hierarchical posterior that is easy to adapt to prior work (Reviewer c1xJ).
>
> We are revising our manuscript to incorporate all reviewers’ feedback. Key updates include:
>
> 1. Clarify the paper’s significance (a framework for multimodal generation and structured representation learning) and originality (the first exact implementation of Hölder pooling with the best quality–coherence trade-off in the current SOTA) (Reviewer c1xJ).
>
> 2. Elaborate on how hierarchical inference encourages disentanglement (Reviewer qjBk), and add CelebAMask-HQ results showing our model outperforms SOTA on generative quality (Reviewer c1xJ).
>
> 3. Add batch runtime analysis showing that, even in the worst-case setting, our method requires only about $1.8\times$ the time of MMVAE-based baselines while effectively pushing the Pareto front (Reviewer Lmmc).
>
> We would greatly appreciate your reconsideration of the score in light of these improvements and would be happy to provide further clarification if needed. Below, we address your specific concerns.
>
> ---
>
> **CUBICC clustering results**
>
> To compare Hölder+ and Hölder++ with MMVAE+ and MMVAE++ under the same standard-prior setting, we run multivariate Hotelling’s $T^2$ tests for image, caption, and joint representations across 10 seeds, **using a significance level of 0.05**. For MMVAE+ vs. Hölder+, the differences are not statistically significant for image ($p=0.503$), caption ($p=0.192$), or joint ($p=0.210$) representations. For MMVAE++ vs. Hölder++, while the joint representation shows a marginal difference ($p=0.066$), the image ($p=0.681$) and caption ($p=0.598$) comparisons are not statistically significant. Therefore, *under the standard prior*, we cannot conclude a significant difference between these models. In contrast, *with a mixture prior*, our CHölder+ and CHölder++ achieve better clustering results than CMVAE, with significant gains in the joint representation. We will clarify this in the revised manuscript.
>
> **Computational complexity**
>
> We acknowledge that we do not provide a theoretical convergence analysis showing that Hölder-based models require fewer epochs to converge, and our discussion is only based on empirical results. We reported average batch training time on **PolyMNIST ($M=5$)** since it has the largest number of modalities in our experiments. For completeness, we additionally report the same metric in the table below for **MNIST-SVHN ($M=2$)** and **CelebAMask-HQ ($M=3$) $^{(1)}$**, where $K$ is the number of samples for ELBO estimation. The results, together with Table 7, show that the runtime gap increases with the number of modalities: for $M=2$, the difference is relatively small, while for $M=3$ and $M=5$ the gap is around $1.8$ times when comparing Hölder vs. MMVAE and Hölder+ vs. MMVAE+. *Thus, we do not observe a quadratic increase in training time as $M$ grows in our benchmarks*; empirically, this suggests that the pairwise term is not the main source of complexity, and can be masked by other factors such as latent dimensionality and model architecture. We will revise the paper to state this result only for the settings we tested ($M \leq 5$). We are not currently aware of an alternative formulation or estimator that would make this computation more efficient, and we leave this for future work.
>
> *Remark.* Our empirical results show that the proposed approach effectively pushes the Pareto front, achieving a better quality–coherence trade-off (see Figs. 2 and 4 in the manuscript). This improvement comes at the cost of increased computational complexity due to the pairwise term, which also contributes to the gains of Hölder+ over SOTA models. In practice, if the number of modalities scales to the point where this additional cost becomes prohibitive, more efficient methods (e.g., MMVAE+ or HELVAE) may be preferable, albeit with a less favorable trade-off between these objectives.
> | |MNIST-SVHN|CelebAMask-HQ|
> |--:|:--:|:--:|
> |HELVAE|0.0303|0.0727|
> |MMVAE ($K=1$)|0.0354|0.1717|
> |Hölder ($K=1$)|0.0457|0.3252|
> |MMVAE+ ($K=1$)|0.0501|0.1940|
> |Hölder+ ($K=1$)|0.0597|0.3510|
> |Hölder++ ($K=1$)|0.0617|0.3518|
> > $^{(1)}$ CelebAMask-HQ is a real-world dataset of 30,000 samples, in which images, masks, and attributes are different modalities describing visual characteristics. In our setting, we use the same backbone and hyperparameters as in [1].
>
> ---
>
> **References**
>
> [1] Wesego et al. Score-Based Multimodal Autoencoder, TMLR 2024.

---

> > ### Author Rebuttal · Reviewer_Lmmc · 2026-04-04
> >
> > Thank the authors for the detailed rebuttal. The authors have fully addressed my concerns. Given the authors' responses and the additional experimental data, I raised my score.

---

> > > ### Author Response · Authors · 2026-04-07
> > >
> > > We sincerely thank the reviewer for reconsidering the paper and for raising the score. We are glad that our response helped address the concerns.
> > >
> > > In light of the comments, we will revise the manuscript to (i) include additional statistical significance tests for the clustering analysis, (ii) add computational complexity results for more datasets and clarify that the claim applies only to the tested settings ($M \leq 5$), and (iii) expand the discussion of limitations and future work.
> > >
> > > We appreciate the reviewer's helpful and constructive feedback, which has helped us improve the overall quality of the paper.

---

### Decision · Program_Chairs · 2026-04-30

**Decision:**

Accept (regular)

**Comment:**

The paper proposes a novel multimodal VAE based on an exact implementation of Holder pooling, distinct shared and private representations and hierarchical inference to further enhance the disentanglement between them. The paper has received 3 reviews with reviewers unanimously recommending acceptance, with ratings (4,4,5).

The reviewers find the paper to be well written and easy to read. The problem setting is well motivated and related to the past art. The technical core is seen to be strong, and mathematically sound. It principally handles quality-coherence tradeoff, exact pooled-posterior decomposition enabling tractable multimodal inference, and hierarchical posteriors easy to adapt to past work. The experiments are seen to validate the method by producing better latent organization and shared representations for downstream tasks. Each of the three proposed improvements are shown to demonstrate benefits.

The initial reviewer concerns related to (a) novelty and significance, (b) evaluation of 1 relatively small dataset, (d) performance improvements over the past work marginal or mixed, (d) per-batch quadratic scaling in the number of modalities M, (e) hierarchical inference to enforce disentanglement and some other clarifications. The author-reviewer discussion phase resolved almost all reviewer concerns as acknowledged by the reviewers. The authors have promised to revise the manuscript to incorporate all reviewers’ feedback.

Overall, I find the paper to be technically sound and making a promising contribution with a potential for high impact although the current experimental validation seems to be at a proof of concept level.